# Serotonergic modulation of vigilance states in zebrafish and mice

Yang Zhao[1,2,5], Chun-Xiao Huang[1,2,5], Yiming Gu[1,2], Yacong Zhao[1,2], Wenjie Ren[1,2], Yutong Wang[1,2], Jinjin Chen[1,2], Na N. Guan [1,2,3,4,6] ✉ & Jianren Song [1,2,3,4,6] ✉

Vigilance refers to being alertly watchful or paying sustained attention to avoid potential threats. Animals in vigilance states reduce locomotion and have an enhanced sensitivity to aversive stimuli so as to react quickly to dangers. Here we report that an unconventional 5-HT driven mechanism operating at neural circuit level which shapes the internal state underlying vigilance behavior in zebrafish and male mice. The neural signature of internal vigilance state was characterized by persistent low-frequency high-amplitude neuronal synchrony in zebrafish dorsal pallium and mice prefrontal cortex. The neuronal synchronization underlying vigilance was dependent on intense release of 5-HT induced by persistent activation of either DRN 5-HT neuron or local 5-HT axon terminals in related brain regions via activation of 5-HTR7. Thus, we identify a mechanism of vigilance behavior across species that illustrates the interplay between neuromodulators and neural circuits necessary to shape behavior states.

The overt behavior of animals is orchestrated by variable internal brain states, in which neural dynamics have profound impact on sensation, cognition, and action[1-5]. Awake animals dynamically adjust their behavioral states as circumstances demand, which requires the brain to generate appropriate internal states that precede behavior change[4,6]. A specific internal state instructs the brain selectively to enhance the sensitivity of certain sensory inputs, while neglect the others[1,7-9]. Neural signature of internal states can be explicitly identified by examining patterns of neuronal activity in cortex or other parts of CNS in many species[2,4,5,7,10,11].

To successfully detect or escape a predator, animals usually maintain a vigilance state, during which they reduce locomotion or even stay stationary while being alertly watchful and attentive to potential threats[7,12,13]. Animals in a state of vigilance have an enhanced sensitivity to aversive stimuli so as to react quickly to predation or dangers and avoid detection by not moving. This may have adverse effects on forging efficiency and group size but is vital for survival[12,13].

The emergence of vigilance behavior is usually triggered by either perceived external threats or internal self-cognition based on experience[7,14], and must be precisely controlled by a corresponding internal brain state, which is maintained until relief of threats. To date, the neural signature representing such an internal brain state as well as its central control mechanism are still unknown. Given the universality of vigilance behavior, it is to be expected that this mechanism will operate in many species[1,2].

5-HT neurons in dorsal raphe nucleus (DRN) provide the major ascending serotonin tone to the forebrain by anatomically broad axon-projection[15-19] and contribute to shape behavior states[3,5,9,20-25]. Functional diversity of 5-HT neurons[15,17,26,27] and spatial distribution and temporal activation of postsynaptic receptors[22,28] should be considered together when evaluating the complex functions of 5-HT system in physiology and behavior[15,22,29,30]. However, it remains unclear how the DRN 5-HT system modulates the pattern of neural dynamics in order to control internal brain states. During wakefulness in mice,

[1]Shanghai Key Laboratory of Anesthesiology and Brain Functional Modulation, Clinical Research Center for Anesthesiology and Perioperative Medicine, Translational Research Institute of Brain and Brain-Like Intelligence, Shanghai Fourth People's Hospital, School of Medicine, Tongji University, Shanghai 200434, China. [2]Clinical Center for Brain and Spinal Cord Research, Tongji University, 200092 Shanghai, China. [3]Frontiers Science Center for Intelligent Autonomous Systems, Tongji University, Shanghai, China. [4]Department of Neuroscience, Karolinska Institutet, 171 77 Stockholm, Sweden. [5]These authors contributed equally: Yang Zhao, Chun-Xiao Huang. [6]These authors jointly supervised this work: Na N Guan, Jianren Song. ✉e-mail: naguan@tongji.edu.cn; song.jianren@tongji.edu.cn

activation of the ascending norepinephrine system can promote an internal state transition from a synchronized state characterized by low-frequency fluctuation in a stationary animal to a desynchronized state with suppressed low-frequency fluctuation in a moving animal[4,6,11]. Studies showed that activation of DRN 5-HT neurons functions in the opposite way, changing an actively moving state into a locomotor-reduced or even stationary state when zebrafish and mice were presented with a variety of behavior challenges[17,23,31–34]. Whether DRN 5-HT neurons and the receptor-specific mechanism could be responsible for generating the internal brain state that underlies vigilance behavior requires further investigations.

Here we present evidence that an unconventional 5-HT-driven innate mechanism operating at circuit-level shapes the internal brain state underlying vigilance behavior in both zebrafish and mice. Persistent and intense activation of DRN 5-HT neurons by a conspecific alarm substance or genetic manipulation converted an internal desynchronized state of active locomotion into a persistent synchronized state of reduced locomotion or stationary behavior. The intense release of 5-HT in the central zone of dorsal pallium (pallial Dc region) of zebrafish and prefrontal cortex of mice generated and maintained the internal vigilance state characterized by low-frequency high-

amplitude neuronal discharge in glutamatergic neurons via targeting of 5-HT7 receptors. Genetic ablation or pharmacological blockade of 5-HT7 receptors abolished the vigilance behavior. Selective optogenetic activation of DRN 5-HT neuron terminals in prefrontal cortex further proved above results and generated the internal vigilance state. Thus, this study provides evidence for an ancestral mechanism controlling vigilance behavior across species.

## Results

### CAS treatment induces a persistent vigilance behavior in zebrafish

Zebrafish was allowed to swim freely in a tank individually to characterize its response to conspecific alarm substance (CAS) treatment by video tracking (Fig. 1a, b). After CAS administration, the zebrafish displayed first a transient high-speed erratic swim and then quickly switched to a persistent reduced locomotion state characterized by a freezing-like behavior staying stationary at tank bottom for about 5 min followed by a slow-speed locomotion for another 20 min (Fig. 1b, c and Supplementary Movie 1). This reduced locomotion state was also observed even if the zebrafish was placed into a new tank filled with fresh water but did not occur when water was administrated as a

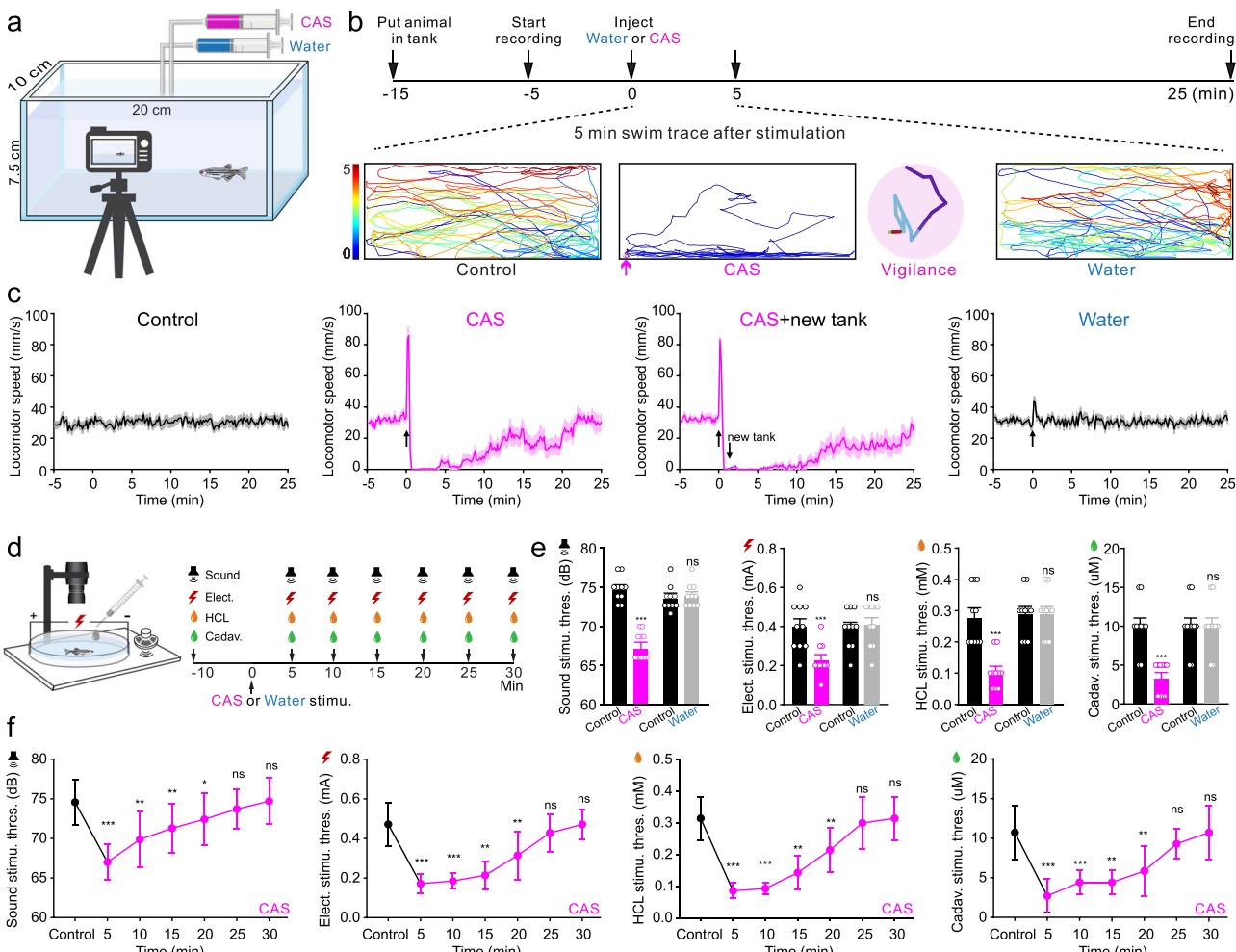

**Fig. 1 | CAS reliably induces a persistent vigilance behavior with increased sensitivity to aversive stimuli in zebrafish. a, b** Illustration of experiment setup and procedures. Bottom panels show typical 5 min swimming patterns before and upon water or CAS addition. **c** Statistics of locomotor speed analysis of 30-min video recording in four groups. The upward arrow indicates injection of CAS or water and the downward arrow indicates transfer of the zebrafish to a new tank with fresh water after CAS injection. Each graph represents the average trace of 10 fish. **d** Illustration of the design and timeline for detecting response thresholds to four

aversive stimuli: sound, electric shock, HCl, and cadaverine five min after prior CAS treatment. **e** Comparison of mean response thresholds to aversive stimuli before and 5 min after CAS or water treatment. *N* = 10 fish in each group. **f** The persistent change of response thresholds to each stimulus after CAS treatment for 30 min. *N* = 6 fish in each group. All data are presented as mean ± SEM. *$P < 0.05$, **$P < 0.01$, ***$P < 0.001$; ns denotes no significant difference. For detailed statistics, see Supplementary Table 2. Source data are provided as a Source Data file.

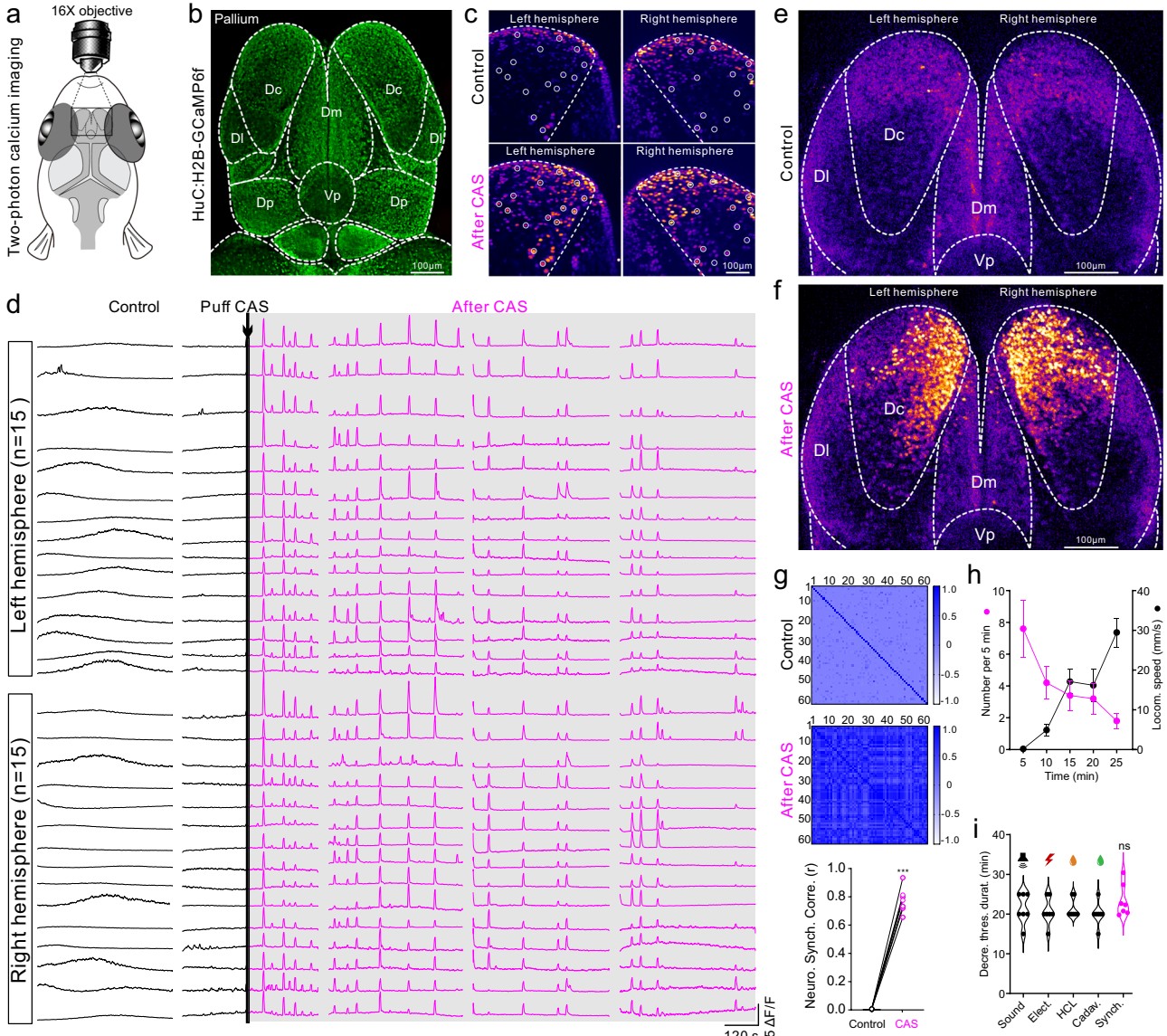

**Fig. 2 | The neural signature of vigilance state is characterized by low-frequency high-amplitude neuronal synchrony in Dc region of zebrafish. a** Illustration of two-photon calcium imaging of zebrafish dorsal pallium. **b** Image stack showing divisions of zebrafish dorsal pallium in the *HuC:H2B-GCaMP6f* fish line. Dc, Dm, Dl, and Dp standing for central, medial, lateral, and posterior zones of dorsal pallium; Vp representing ventral pallium. **c** Calcium imaging with cellular resolution in Dc region before and after CAS treatment. Each white circle indicates a single neuron. **d** Analyzed traces of calcium oscillation of 30 neurons in both left and right Dc region as indicated in (**c**) for 30 min. After CAS treatment, these neurons display synchronized calcium oscillations. **e, f** Stack image of 6-layer volume-scan of the whole dorsal pallium in zebrafish before and after CAS treatment showing the spatiotemporal pattern of neuronal activity. **g** Paired Pearson correlation analysis of calcium oscillation between neurons (*n* = 60) in Dc region before and after CAS treatment, and statistics of correlation coefficient for seven fish. The scale bar indicates Pearson *r* value. **h** Continuous decrease of synchronized oscillation numbers is accompanied with a persistent increase of locomotor speed in zebrafish after CAS treatment. *N* = 5 fish. **i** Duration of decreased response thresholds to four aversive stimuli and the duration of neuronal synchrony. *N* = 7 fish. Images in (**b**, **c**), **e**, **f** represents results from six independent experiments. All data are presented as mean ± SEM. ***P* < 0.001; ns denotes no significant difference. For detailed statistics, see Supplementary Table 2. Source data are provided as a Source Data file.

negative control (Fig. 1b, c). To determine that the locomotor-reduced state is a truly petrified[35–37] or highly vigilant state sensitized for the detection of potential dangers[23], we measured the response threshold to aversive stimuli including sound, electric shock (Elect.), HCL, and cadaverine (Cadav.) in water- or CAS-treated fish (Fig. 1d and Supplementary Fig. 1). The zebrafish receiving CAS displayed significantly decreased thresholds to the above aversive stimuli, which slowly returned to the control level (Fig. 1d–f and Supplementary Fig. 1). No significant changes were observed in water-treated fish.

CAS treatment serving as an alarm signal reliably generates a persistent vigilance behavior state characterized by being motionless

but simultaneously alert to aversive stimuli in zebrafish. This provides a robust model system to study the neural basis of internal brain state underlying vigilance behavior.

## Neural signature of internal vigilance state

To identify the pattern of neural dynamics underlying vigilance behavior, we examined neuronal activity of dorsal pallium by volume scanning of intracellular calcium signal with two-photon microscope in the *Tg(HuC:H2B-GCaMP6f)* fish line with pan-neuronal nucleus-localized GCaMP6f (Fig. 2a, b). The volumetric imaging of the whole dorsal pallium in six layers revealed spatiotemporal pattern of neuronal

activity. After CAS treatment, a large proportion of neurons in both hemispheres of dorsal pallium immediately switched calcium dynamic pattern from a desynchronized state to a persistent synchronized state displaying low-frequency high-amplitude oscillations (Fig. 2c, d and Supplementary Movie 2). A volumetric stack of six imaged layers showed that the vast majority of synchronized neurons were located in the central zone of dorsal pallium (pallial Dc region, homologous to mammalian cortex)[38] (Fig. 2e, f and Supplementary Fig. 2). High synchrony of neuronal oscillation after CAS treatment was confirmed statistically by a cross-correlation coefficient between individual neurons (Fig. 2g). Oscillation numbers in the synchronized state decreased as locomotor speed increased (Fig. 2h, i). Moreover, the duration of synchronized neuronal oscillation was similar to the period of persistently decreased threshold of aversive stimuli (Fig. 2h, i). These data suggest that CAS treatment initiates and maintains an internal synchronized state of neuronal dynamics underlying vigilance behavior in zebrafish.

### Persistent activation of DRN 5-HT neurons shapes the vigilance state

The involvement of DRN 5-HT neurons in locomotor reduction[23,32] prompted us to study their role in vigilance. We examined the function of DRN 5-HT neurons in controlling the internal vigilance state by two-photon imaging in the *Tg(tph2: Gal4;UAS:GCaMP6f)* fish line (Fig. 3a). CAS treatment induced an immediate and intense activation of nearly all DRN 5-HT neurons that lasted for more than 120 s (Fig. 3b–d and Supplementary Fig. 3a). This was confirmed by extracellular recordings of individual 5-HT neurons persistently discharging at a much higher frequency in *Tg(tph2: GFP)* line (Fig. 3e–h). These findings suggest a persistent and intense activation of DRN 5-HT neurons induced by CAS treatment that closely correlates with the advent of the internal vigilance state.

To determine if activation of DRN 5-HT neurons could contribute to generation of the internal vigilance state, we first confirmed the projection pattern of these neurons to the Dc region using photo-convertible *Tg(tph2: kaede)* fish line (Fig. 3i). Kaede photo-conversion of axon terminals specifically targeted to the Dc region converted the green fluorescence to red in about 82% of DRN 5-HT neuron soma, confirming the intense projection of these neurons into the Dc region (Fig. 3j–l). Release of 5-HT in dorsal pallium after CAS treatment was also measured using liquid chromatography-mass spectrometry (LC–MS, Fig. 3m). We found that the concentration of 5-HT in the dorsal pallium was robustly increased at 5 min after CAS stimulation and then after reuptake or metabolized returning to control levels after 1 h (Fig. 3n, o). We next generated a *tph2* (*tryptophan hydroxylase 2*) null mutant line *tph2*[-/-] *Tg(HuC:H2B-GCaMP6f)* to evaluate the function of 5-HT neurons in generating and maintaining neuronal synchrony of internal vigilance state by genetic ablation of these neurons. Compared with their wild-type siblings, CAS treatment failed to induce neuronal synchrony in the Dc region of the *tph2*[-/-] line, while exogenous application of 5-HT restored that synchrony (Fig. 3p–s). Consistent with the loss of neuronal synchrony, the *tph2*[-/-] fish did not display any characteristics of the vigilance behavior after CAS treatment, while their siblings responded normally (Fig. 3t). Consistently, chemogenetic ablation of raphe 5-HT neurons using the *Tg(tph2: Gal4;UAS:nfsB-mCherry)* crossed with the *Tg(HuC:H2B-GCaMP6f)*[39] further confirmed that loose of 5-HT neurons abolished both CAS-induced neuronal synchrony and vigilance behavior (Supplementary Fig. 3). The further experiments demonstrated that the erratic swim (EMG) occurred first, then the intense activation of DRN 5-HT neurons followed and finally the neuronal synchrony in Dc appeared (Supplementary Fig. 3b).

In conclusion, CAS treatment induced persistent and intense activation of raphe 5-HT neurons, resulting in strong release of 5-HT in the Dc region through long-projecting axon. The high concentration of 5-HT in the Dc region generated and maintained the internal

synchronized state with low-frequency high-amplitude neuronal oscillation as well as the vigilance behavior. As the decrease of 5-HT concentration in the Dc region, the number of synchronized neuronal oscillation subsequently declined and the threshold of aversive stimuli gradually returned to the control level.

### The synchronized neurons in Dc are *5-HT7a*-expressing neurons

We determined to identify the receptor-specific mechanism allowing 5-HT to produce the neuronal synchronization in Dc region. We performed whole-cell recording in both synchronized and non-synchronized neurons in the *Tg(HuC:H2B-GCaMP6f)* fish line under two-photon microscope, which enabled us to correlate single neuron activity with synchronized oscillation of neuronal population (Fig. 4a, b). In the recorded synchronized neurons, CAS treatment switched the spontaneous firing pattern into a bursting firing pattern and a post-inhibitory rebound consistently occurred at the onset of each synchronized oscillation (Fig. 4c, e), which was not observed in the non-synchronized neurons (Fig. 4d, f and Supplementary Fig. 4a, b). We further performed single-cell RNA-sequencing in the two identified neuron types (Fig. 4g), which revealed that the synchronized neurons expressed much higher level of glutamate transporter genes (*slc17a7a, slc17a6a*), while non-synchronized neurons showed significantly higher expression of *gad1b*, suggesting that synchronized neurons are primarily glutamatergic, while non-synchronized neurons are mainly GABAergic (Fig. 4h). Among the 5-HT receptor genes found, *htr7a* was exclusively expressed in the synchronized neurons (Fig. 4h).

To validate the RNA-sequencing results, the *htr7a*[-/-] Tg(*vglut2a:DsRed*) fish line was generated (Fig. 4j). After CAS treatment all *vglut2a*-positive neurons recorded by whole-cell patch-clamp in the Dc region of wild-type animals changed their discharge pattern from spontaneous to persistent burst firing, during which the occurrence of post-inhibitory rebound was constantly observed (Fig. 4j). The bursting firing pattern as well as the post-inhibitory rebound disappeared in animals either when preincubated with a selective 5-HT7 receptor antagonist SB-269970 (Fig. 4j) or when using the *htr7a*[-/-] Tg(*vglut2a:DsRed*) animal (Fig. 4j). The same results were obtained by exogenous application of 5-HT instead of CAS treatment (Supplementary Fig. 4c). We never found any whole-cell recorded GABAergic neurons changed its firing pattern after CAS administration (Fig. 4k). This confirmed that the neurons displaying low-frequency high-amplitude synchronized oscillation are *htr7a*-expressing glutamatergic ones.

The neuronal synchrony underlying the internal vigilance state in the Dc region was also prevented by preincubation with a selective 5-HT7 receptor antagonist or when using *htr7a*[-/-] Tg(*HuC:H2B-GCaMP6f*) fish line (Fig. 4l–o). Exogenous application of 5-HT failed to restore the neuronal synchrony due to the absence of 5-HT7 receptors in the *htr7a*[-/-] Tg(*HuC:H2B-GCaMP6f*) fish (Supplementary Fig. 4d). The behavior analysis showed that CAS treatment failed to generate and maintain the vigilance behavior in the *htr7a*[-/-] fish (Fig. 4p and Supplementary Movie 3). 5-HT7 receptor activation was previously shown to inhibit SK channels and thus increase neuronal excitability[40–42]. We found that the *kcnn3* gene encoding SK channels was enriched in the synchronized neurons (Fig. 4h). Preincubation with a SK channel activator abolished the bursting firing pattern as well as the post-inhibitory rebound in glutamatergic neurons in Dc region induced by CAS treatment or application of 5-HT (Supplementary Fig. 4e, f), suggesting that 5-HT7-receptor activation triggers inhibition of SK channels. An additional 5-HT receptor mutation fish line was generated and these fish reliably displayed vigilance behavior after CAS treatment (Supplementary Fig. 4g–j).

In brief, the intense release of 5-HT induced by CAS treatment activates 5-HT7 receptors on glutamatergic neurons in the Dc region, which results in altered neural circuit activity pattern and generation of the synchronized state and vigilance behavior.

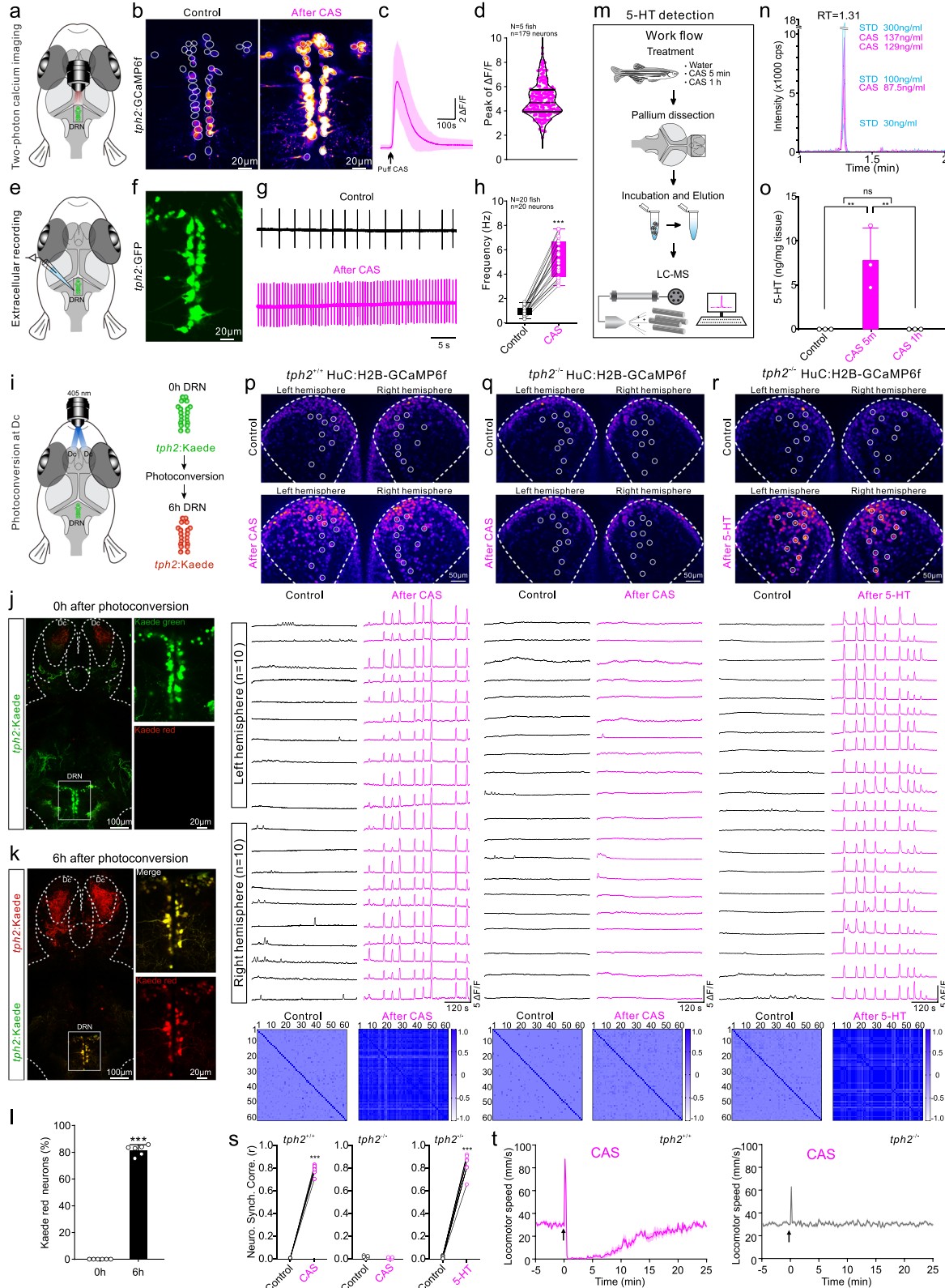

## An innate mechanism generating the internal vigilance state

We next examined if persistent and intense activation of DRN 5-HT neurons is an innate mechanism to generate and maintain the synchronized state and vigilance behavior using the *Tg(tph2:Gal4;5 × UAS:ChR2(H134R)-mCherry)* fish line (Fig. 5a). A lasting blue-light stimulation for 120 s restricted to the DRN region activated by CAS treatment, triggered a persistent and intense activation of DRN

5-HT neurons (Fig. 5b), which in turn induced burst firing and post-inhibitory rebound in the whole-cell recorded glutamatergic neurons of the Dc region (Fig. 5c, d) in *Tg(vglut2a:DsRed)* (red fluorescent protein was driven by the promoter of vesicular glutamate transporter gene) crossed with *Tg(tph2:Gal4;5 × UAS:ChR2(H134R)-mCherry)* fish line. Furthermore, optogenetic stimulation consistently generated and maintained a vigilance-like behavior in

**Fig. 3 | CAS treatment generates and maintains internal synchronized state and vigilance behavior through persistent and intense activation of DRN 5-HT neurons in zebrafish. a** Configuration for performing two-photon calcium imaging on DRN 5-HT neurons in the *tph2:GCaMP6f*. **b** Typical images illustrating calcium change after CAS treatment. **c** Averaged calcium changes of 40 neurons in response to CAS treatment in one fish. **d** Statistical quantification of calcium activity in 179 neurons from five fish. **e, f** Configuration of extracellular recordings from DRN 5-HT neurons using *tph2:GFP*. **g, h** Typical extracellular recording from one neuron (**g**) and statistic of firing frequency of 20 neurons from 20 fish (**h**) before and after CAS treatment. Center of boxplots is the median, upper and lower limits of boxes are the first and third quartile respectively and the upper and lower limits are the maxima and minima. **i** Illustration of photo-conversion of DRN 5-HT neurons in *tph2:Kaede*. **j, k** Fluorescent images showing converted Kaede protein from 5-HT projections in Dc region transported to DRN 5-HT soma (enlarged white box) after 6 h. **l** Mean data of conversion percentage. **m** LC–MS workflow for detection of 5-HT concentration in dorsal pallium after CAS treatment. **n** Mass spectrometry chromatograms showing retention time of standard serotonin (blue) and elution samples at 5 min after CAS treatment (pink). **o** Mean concentration of 5-HT in dorsal pallium at 5 min, 1 h after CAS treatment, and in control group. **p–r** Two-photon calcium images and the analyzed data showing CAS-induced neuronal synchronization in Dc region in *tph2⁺/⁺ HuC:H2B-GCaMP6f* (**p**). The neuronal synchronization disappeared in *tph2⁻/⁻ HuC:H2B-GCaMP6f* after CAS treatment (**q**), which is restored by exogenous application of 5-HT (**r**). **s** Illustration of the correlation coefficient for the data presented in (**p–r**). **t** Quantification of locomotor speed of *tph2⁺/⁺* and *tph2⁻/⁻* fish in response to CAS treatment in 30-min recording. Images in b, j-k represents results from six independent experiments. All data are presented as mean ± SEM. \*\**P* < 0.01, \*\*\**P* < 0.001; ns denotes no significant difference. For detailed statistics, see Supplementary Table 2. Source data are provided as a Source Data file.

*Tg(tph2:Gal4;5 × UAS:ChR2(H134R)-mCherry)* fish line, but not in *Tg(tph2:Gal4;5 × UAS: mCherry)* fish line used as a negative control group (Fig. 5e–g, Supplementary Movie 4). The duration and depth of the vigilance-like behavior state could be modulated by varying the duration or intensity of the applied blue light (Fig. 5g and Supplementary Fig. 5). The vigilance-like behavior induced by optogenetic stimulation was further confirmed to be vigilance behavior by the presence of decreased thresholds to aversive stimuli (Fig. 5h–j). These results suggest that persistent and intense activation of DRN 5-HT neurons serves as an innate mechanism to generate and maintain the internal brain state as well as vigilance behavior in zebrafish.

## Mice share the same innate mechanism for vigilance with zebrafish

DRN 5-HT neurons are known to project intensely to cortex in mice[16–19,43]. To determine whether the innate mechanism controlling vigilance behavior revealed in zebrafish was species-specific or a general mechanism also existing in mammal, we used optogenetic activation of DRN 5-HT neurons in mice to evaluate their internal state and behavior performance. Multi-channel electrodes were implanted in mice prefrontal cortex (PFC) for simultaneous recording of single-neuron activity and EEG signals while performing blue-light stimulation and behavior tests (Fig. 6a and Supplementary Fig. 6a). We performed viral injection of pAAV-Ef1a-DIO-hChR2(H134R)-mCherry or pAAV-Ef1a-DIO-mCherry-WPRE in DRN using Slc6a4-Cre mice[44], in which the serotonin transporter (SERT) promoter specifically drives Cre expression in 5-HT neurons (Fig. 6a, b and Supplementary Fig. 6a, b). Post hoc histological analyses confirmed hChR2-mCherry or mCherry expression restricted to the DRN 5-HT neurons (Fig. 6b, c and Supplementary Fig. 6b–d). We found lasting blue-light stimulation for 120 s induced persistent and intense activation of the recorded hChR2-positive 5-HT neurons in DRN, while pulsed stimulation (10 Hz or 20 Hz) only produced moderate effects (Fig. 6c and Supplementary Fig. 6c, d). Interestingly, persistent blue-light stimulation for 120 s in free-moving mice exclusively generated and maintained a synchronized state with low-frequency oscillation of EEG as well as a reduction in locomotion (Fig. 6d and Supplementary Movie 5). The dominant power density of cortical EEG switched from alpha band spectra (8–13 Hz) in a desynchronized state to low-frequency delta band spectra (1–4 Hz) in a synchronized state accompanying with locomotor-reduced behavior (Fig. 6d and Supplementary Movie 5). The internal synchronized state and reduced locomotion were not observed in the mice given intraperitoneal injection of a 5-HT7 receptor antagonist (Fig. 6e) or in the virus control mice (Fig. 6f). The power of these effects was largely influenced by intensity and duration of the applied blue light (Fig. 6d and Supplementary Fig. 6e). Pulsed blue-light stimulation failed to change EEG power spectra and only increased locomotor activity, which previously was considered as anxiety behavior[26,31] (Supplementary Fig. 6f, g and Supplementary Movie 5). Single-unit recording of neurons in the prefrontal cortex revealed that excitatory neurons identified by

spike-shape sorting displayed a burst-firing pattern in phase with the low-frequency high-amplitude EEG oscillation generated by persistent blue-light stimulation (Fig. 6g–i). The generation and maintenance of such an internal brain state determined the behavior performance. The displayed locomotor-reduced state in mice triggered by persistent blue-light stimulation, was confirmed to be vigilance behavior by the long-lasting decrease in threshold to aversive stimulation (Fig. 6j–l) and mice hiding themselves for significantly longer times in the shelter upon exposure to the aversive stimulation compared to mice in the control group (Fig. 6m–o). Persistent blue-light stimulation failed to generate vigilance behavior in the mice receiving intraventricular injection of the 5-HT7 receptor antagonist SB-269970 in prefrontal cortex or in the virus control mice (Fig. 6j–o). This suggests that persistent and intense activation of DRN 5-HT neurons reliably generates and maintains an internal synchronized state through a robust release of 5-HT acting on 5-HT7 receptor-expressing excitatory neurons in prefrontal cortex.

## Activation of DRN 5-HT neurons generates vigilance state in mice

The 5-HT-driven vigilance behavior state was confirmed by additional experiments using chemogenetic activation of DRN 5-HT neurons or exogenous administration of 5-HT. The DRN of Slc6a4-Cre mice was infected with a viral vector containing Cre-dependent hM3d(Gq), whose specific expression in DRN 5-HT neurons was confirmed by post hoc staining (Supplementary Fig. 7a, b). Application of clozapine N-oxide (CNO) in the brain slice induced persistent and intense activation of the recorded DRN 5-HT neurons in the hM3d(Gq)-expressing mice, but not in the purely mCherry-expressing mice as the control virus group (Supplementary Fig. 7c, d). CNO treatment via intraperitoneal injection led to a significantly more pronounced decrease in locomotion speed in the hM3d(Gq)-expressing mice (Supplementary Fig. 7e) compared with the control virus group (Supplementary Fig. 7f). The locomotor-reduced state was confirmed to be the vigilance behavior state with the additional findings of a prolonged decreased threshold to aversive stimuli as well as the significantly longer duration of hiding in the shelter (Supplementary Fig. 7g–l). Exogenous application of 5-HT by intraventricular injection into the prefrontal cortex confirmed the above results and consistently generated and maintained the vigilance behavior, which were diminished by simultaneous intraventricular co-injection of a 5-HT7 receptor antagonist (Supplementary Fig. 8). This further highlights the crucial role of 5-HT and 5-HT7 receptors in generating and maintaining vigilance behavior state in both zebrafish and mice. Taken together, these findings reveal an evolutionarily conserved 5-HT-driven mechanism that can reliably generate and maintain the internal synchronized state and vigilance behavior in both zebrafish and mice.

## The projection of DRN 5-HT neurons to PFC encode vigilance state in mice

To further investigate whether projection of DRN 5-HT neurons to PFC was critical to generate vigilance state in mice, we implanted bilateral

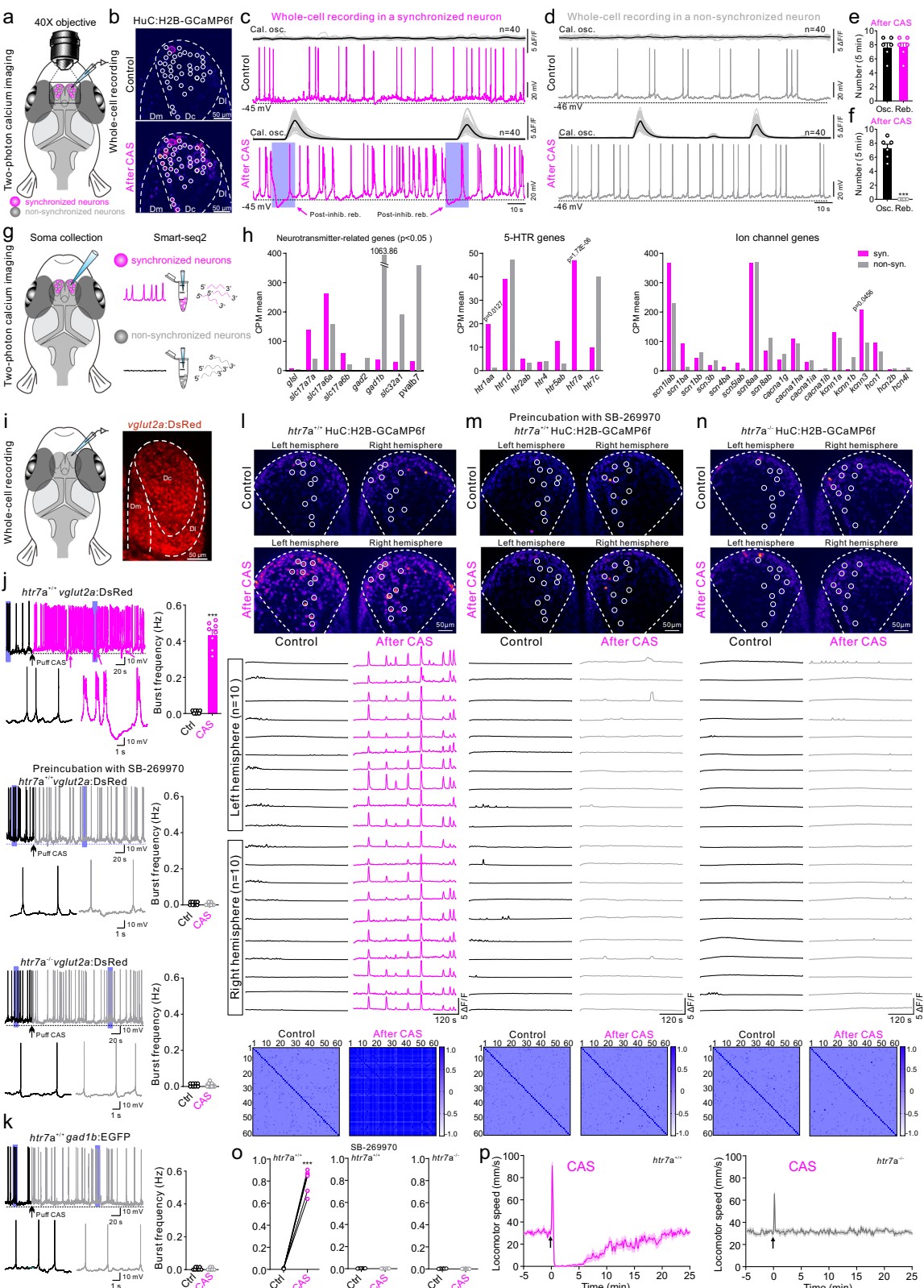

optical fibers and photo-stimulated prefrontal cortex in Slc6a4-Cre mice with viral injection of pAAV-Ef1a-DIO-hChR2(H134R)-mCherry or pAAV-Ef1a-DIO-mCherry-WPRE as a virus control group in DRN (Fig. 7a). The expression of hChR2-mCherry was observed in DRN 5-HT neuron axon terminals in the PFC region (Fig. 7b). Persistent blue-light stimulation of the hChR2-positive terminals in PFC bilaterally for 120 s in free-moving mice specifically generated and maintained an internal

synchronized state with low-frequency oscillation of EEG accompanied by locomotor reduction (Fig. 7c, d). The dominant power density of cortical EEG was changed from alpha band spectra (8–13 Hz) in the desynchronized state to a low-frequency delta band spectrum (1–4 Hz) in the synchronized state accompanying locomotor-reduced behavior state (Fig. 7c). Neither an internal synchronized state nor locomotor reduction were observed in the virus control group (Fig. 7e, f). These

**Fig. 4 | The released 5-HT generated internal synchronized state through acting on 5-HTR7 of glutamatergic neurons in Dc region. a** Experimental setup combing two-photon calcium imaging with whole-cell recording of synchronized (pink) or non-synchronized (gray) neurons in Dc region of *HuC:H2B-GCaMP6f*. **b** Calcium imaging showing synchronized neurons (white circles) after CAS treatment and the recorded neuron (pink circle). Image represents results from six independent experiments. **c, d** Whole-cell recording of a synchronized (**c**) or non-synchronized (**d**) neuron along with simultaneous two-photon calcium imaging of 40 neurons in Dc region before and after CAS treatment. The post-inhibitory rebound in membrane potential of the synchronized neurons aligns with the onset of corresponding calcium oscillations. **e, f** Statistic of calcium oscillation and post-inhibitory rebound per 5 min after CAS administration in synchronized (**e**) and non-synchronized (**f**) neurons. **g** Illustration of identification and collection of the synchronized and non-synchronized neurons for RNA-sequencing. **h** Differences in gene expression of neurotransmitters, 5-HT receptors, and ion channels between the synchronized and non-synchronized neurons. **i** Illustration of whole-cell recordings of

glutamatergic neurons in the Dc region using *vglut2a:DsRed*. **j** CAS treatment induced bursting firing and post-inhibitory rebound in glutamatergic neurons in Dc region of *htr7a+/+ vglut2a:DsRed* (upper). Preincubation of 5-HT7 receptor antagonist SB-269970 (0.3 μM, middle) or using *htr7−/− vglut2a:DsRed* (lower) prevented the burst firing and post-inhibitory rebound. The shaded area is enlarged. **k** CAS treatment had no effect on the electrical activity of GABAergic neurons in *htr7a+/+ gad1b:EGFP* fish. **l–n** Two-photon calcium images, the analyzed data and cross-correlation analysis of calcium oscillation between 60 neurons showing neuronal synchronization in Dc region after CAS treatment in *htr7a+/+ HuC:H2B-GCaMP6f* (**l**). Preincubation of SB-269970 prevented CAS-induced neuronal synchronization in (**m**). The neuronal synchronization disappeared in *htr7a−/− HuC:H2B-GCaMP6f* after CAS treatment (**n**). **o** Statistics of correlation coefficient in (**l–n**). Each point represents the mean value of one fish. **p** Quantification of locomotor speed of *htr7a+/+* and *htr7a−/−* fish in response to CAS treatment in 30-min recording. $N = 10$ fish. All data are presented as mean ± SEM. ***$P < 0.001$. For detailed statistics, see Supplementary Table 2. Source data are provided as a Source Data file.

results indicate that direct activation of DRN 5-HT neuron terminals in PFC generates and sustains both an internal synchronized state and vigilance behavior in mice.

## Pupil size persistently increased during vigilance behavior

Variations in pupil size are well known to be coupled with arousal states or phasic activation of DRN 5-HT neurons[4,5,45]. To investigate the pupillary size fluctuation during vigilance behavior, we tracked the changes in pupil size in the head-fixed mice upon optogenetic or chemogenetic activation of DRN 5-HT neurons using viral injection in DRN of Slc6a4-Cre mice (Fig. 8a and Supplementary Fig. 9a). Blue light or CNO application in corresponding virus-infected mice persistently increased the pupil size and keep the pupillary fluctuation at a more pronounced level (Fig. 8b–e and Supplementary Fig. 9b, c), while no significant changes were found in the control virus group (Fig. 8b–e and Supplementary Fig. 9d, e). Persistent optogenetic activation of DRN 5-HT axon terminals in PFC also kept the fluctuation in pupillary size at an elevated level, even though the increased amplitude was not as high as that induced by activation of DRN 5-HT neurons (Supplementary Fig. 10). These findings suggest that the persistent increase in pupil size is an essential part of the 5-HT-driven vigilance behavior state.

## Discussion

Neuromodulators are characterized as the central elements in generating behavior states[1–5,9,20,21,46,47] and known to regulate function of neural circuit[29,30,48,49]. Less is known about how neuromodulators interact with neural circuits to generate internal brain states[1,46]. Here we combined functional imaging, electrophysiology, genetic manipulation, and anatomical tracing to uncover a 5-HT-driven innate mechanism that converted the freely moving state to a locomotor-reduced vigilance state in both zebrafish and mice. During the behavior state transition, the activity pattern of neural circuit in the pallial Dc region of zebrafish or in the prefrontal cortex of mice was transformed from the desynchronized state to the synchronized state displaying low-frequency high-amplitude neuronal oscillation. The neural underpinning of such internal state transformation was subsequently uncovered. An intense release of 5-HT in Dc region of zebrafish or prefrontal cortex of mice targeted the 5-HT7 receptors on glutamatergic neurons, resulting in a change of firing pattern from spontaneous single spike firing to burst firing. The occurrence of post-inhibitory rebounds in the recorded glutamatergic neurons during the internal synchronized state were perfectly aligned with the onset of calcium oscillations of neuronal populations, which was never observed in the internal desynchronized state. Taken together these results conclusively demonstrate the existence of a 5-HT-driven activation of 5-HTR7 in generating and maintaining internal vigilance state

by altering the activity pattern of neural circuit in both zebrafish and mice brain.

In this study, we used the broadest definition of the vigilance as being alert watchful or paying sustained attention to avoid potential threats[7]. Facing with external predation animal usually escape to a safer space to hide themselves and stay still to better detect danger signals[14,23,33,34,50,51]. The self-cognition based vigilance, such as witnessing group companions being attacked by predators, also reminds animals to maintain a vigilance behavior state to avoid similar scenarios[12,13,52]. Until now, the neural basis of vigilance behavior was completely unknown due to the lack of animal models and behavioral paradigms. Here we found that CAS could reliably produce and maintain vigilance behavior and the internal brain state in zebrafish. Taking advantage of this paradigm, we uncovered the neural signature as well as the 5-HT-driven mechanism underlying internal vigilance state in zebrafish and also in mice. Persistent and intense activation of DRN 5-HT neurons could reliably generate and maintain both an internal synchronized state and vigilance behavior in these two animal models. We further revealed that the 5-HT-driven mechanism controlling internal vigilance state was mediated by 5-HT7 receptors in the glutamatergic neurons of zebrafish Dc region and mice prefrontal cortex. The effect of 5-HT7 receptors was mediated through inhibition of SK channels since activation of SK channels prevented the 5-HT-driven synchronized state in zebrafish. Activation of 5-HT7 receptors have been shown to switch the neuronal firing pattern from a spontaneous to a bursting firing pattern via inhibition of SK channels in hippocampus pyramidal neurons[40–42,53] leading to synchronized neuronal activity[41]. In addition, fluctuations in pupil size in mice were persistently increased in response to activation of DRN 5-HT neurons matching up with vigilance behavior. This is consistent with that behavior state transition is usually accompanied with pupillary fluctuation[1,2]. Taken together, these behaviors induced by a range of external and internal cues may represent an innate, ancestral mechanism across species.

Pharmacological and genetic activation of DRN 5-HT neurons were shown to either decrease locomotor activity[23,31] or enhance anxiety-like behavior[39]. The comprehensive characterization combining anatomy, physiology, and function in mice revealed two functionally heterogenous DRN 5-HT subgroups with one projecting to amygdala promoting anxiety-like behavior[17,26] and the other projecting to frontal cortex innervating neurons responsible for locomotor restriction in the face of challenges[17]. Visualization of DRN 5-HT neurons by miniaturized microscopy further revealed that the population activity of these neurons was significantly higher in the face of an external threat than when using neural stimulation in mice[27,54]. Our results in mice showed that persistent and intense stimulation of DRN 5-HT neurons was sufficient to generate and maintain the internal

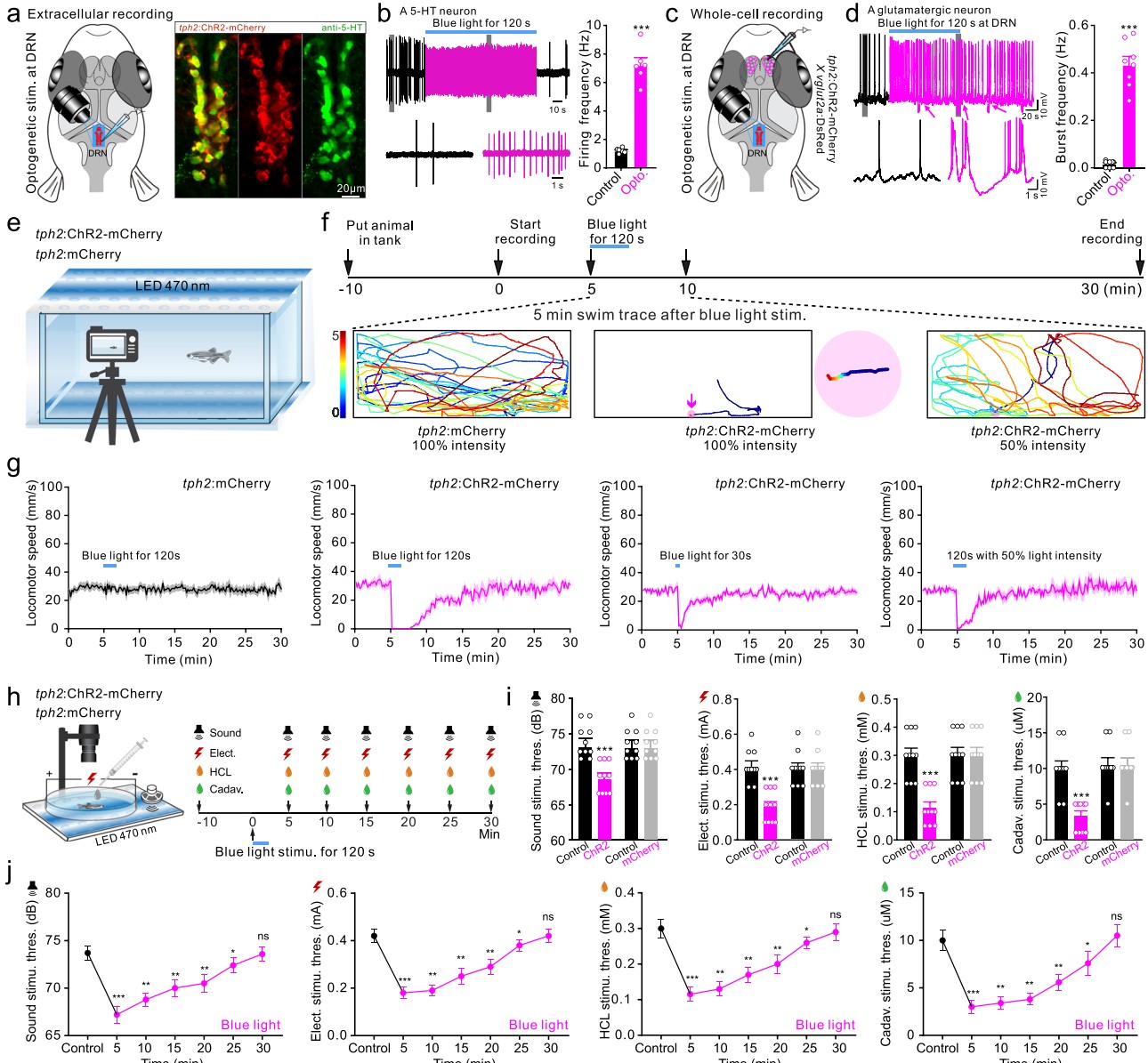

**Fig. 5 | Persistent optogenetic activation of DRN 5-HT neurons generates internal synchronized state and vigilance behavior in zebrafish. a** Illustration of optogenetic stimulation combined with extracellular recording of DRN 5-HT neurons using *tph2:ChR2-mCherry* fish to evaluate activation efficiency (left). Immunofluorescence using anti-serotonin showing DRN 5-HT neurons expressing ChR2-mCherry (right). Image represents results from six independent experiments. **b** Light stimulation for 120 s persistently increased discharge frequency in a DRN 5-HT neuron (left). Mean data of firing frequency of six neurons from six fish before and after optogenetic activation (right). **c** Illustration of optogenetic activation of DRN 5-HT neurons combined with whole-cell recording of glutamatergic neurons in Dc region using *tph2:ChR2-mCherry* crossed with *vglut2a:DsRed* fish. **d** Whole-cell recording from a glutamatergic neuron that displayed burst firing and a post-inhibitory rebound after optogenetic activation of DRN 5-HT neurons (left). Mean data showing burst-firing frequency of eight neurons from eight fish before and after optogenetic activation (right). **e** Optogenetic stimulation setup used for free-swimming zebrafish. **f** Timing of experimental procedures to record behavior

responses to optogenetic activation of DRN 5-HT neurons (upper). Effect of 100% or 50% maximum intensity blue-light stimulation on locomotion at five minutes post stimulation in control and *tph2:ChR2-mCherry* fish (lower). **g** Mean locomotor speed for 30-min video tracking after blue-light stimulation with different duration or intensity in *tph2:mCherry* as control group or *tph2:ChR2-mCherry* fish. 100% light intensity equals 15 mW LED power. $N = 10$ fish in each group. **h** Illustration of the method used to detect the response thresholds to aversive stimuli after optogenetic activation of DRN 5-HT neurons. **i** Mean data shows that the response thresholds to aversive stimuli were markedly reduced in *tph2:ChR2-mCherry* fish 5 min after optogenetic activation of DRN 5-HT neurons. **j** Mean data showing that a reduction in response thresholds to aversive stimuli that endured for more than 20 min after optogenetic stimulation. All data are presented as mean ± SEM. $*P < 0.05$, $**P < 0.01$, $***P < 0.001$; ns denoting no significant difference. For detailed statistics, see Supplementary Table 2. Source data are provided as a Source Data file.

synchronized state and vigilance behavior, while pulsed stimulation induced only anxiety-like behavior[26]. This is likely due to much stronger activation of DRN 5-HT neurons by persistent and intense stimulation than pulsed stimulation that was revealed by whole-cell recording in these neurons in this study. Thus, persistent stimulation

with optogenetics may preferentially promote the function of DRN[5-HT]-frontal-cortex subsystem while pulsed stimulation preferentially promotes the role of DRN[5-HT]-amygdala subsystem, resulting in generation of either vigilance behavior or anxiety-like behavior. Persistent activation of DRN 5-HT neuron axon terminals distributed in the prefrontal

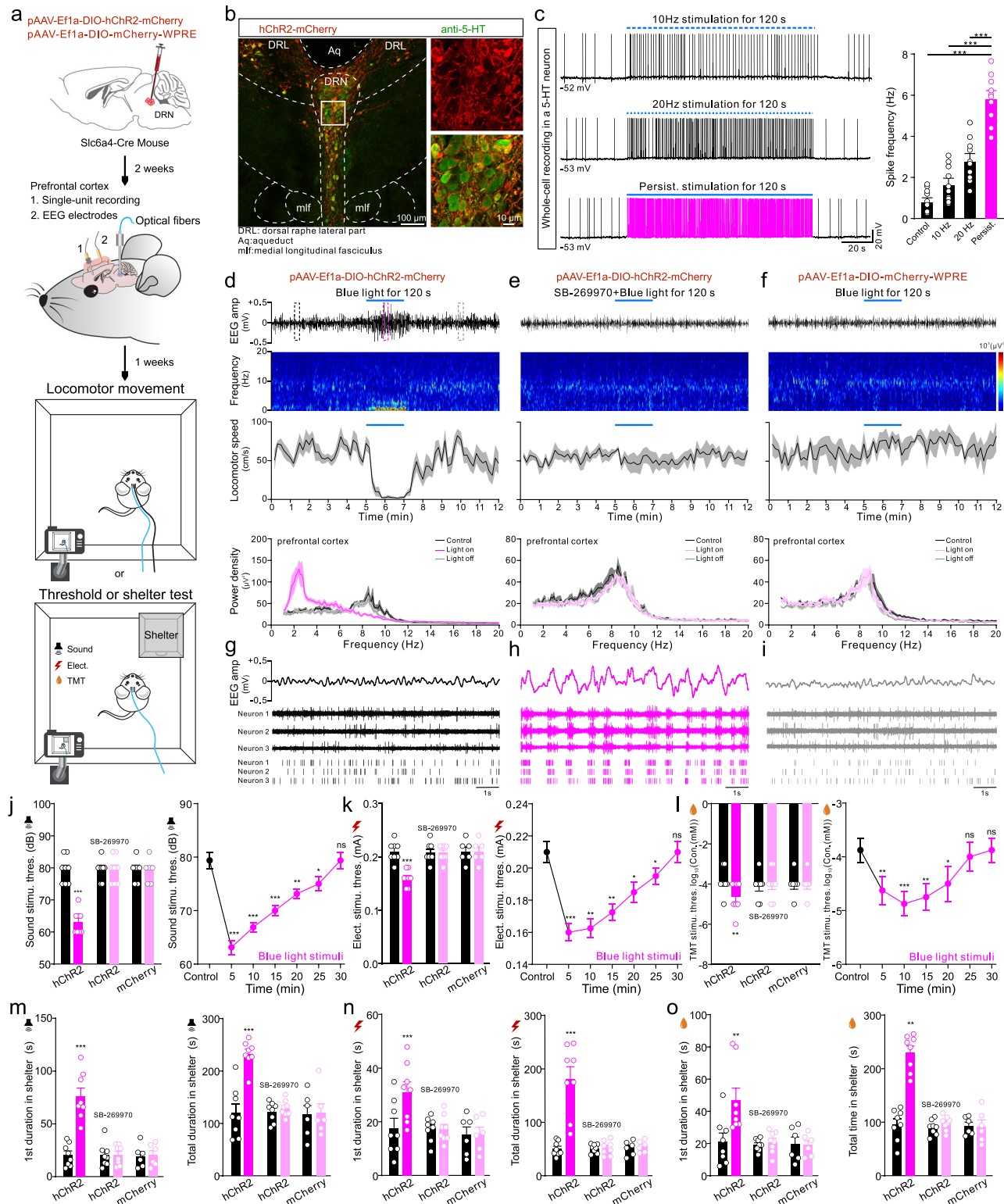

cortex exclusively generated and maintained the vigilance state, suggesting the crucial role of the projection of DRN[5-HT]-PFC subsystem. Even though not all the DRN 5-HT neurons project to Dc region in zebrafish (82% in this study) and prefrontal cortex in mice[16–19,43,55], generation of internal vigilance state requires to recruit the majority of DRN 5-HT neurons revealed in this study. Our results do not exclude the possible contribution of projection of DRN 5-HT neurons to other brain regions in the generation of vigilance behavior. It seems that the subgroups of DRN 5-HT neurons sometimes may work in a

synergistical way to accomplish the specific tasks. Further investigation using more specific genetic tools is required to clarify this.

In summary, we uncovered an ancestral 5-HT-driven mechanism controlling internal synchronized state and vigilance behavior across species. A persistent and intense activation of DRN 5-HT neurons generates low-frequency high-amplitude oscillatory activity pattern of neural circuit in cortex-like region or cortex by an intense release of 5-HT that acts through the 5-HT7 receptor in zebrafish and mice.

**Fig. 6 | Persistent optogenetic activation of DRN 5-HT neurons induce internal synchronized state and vigilance behavior in mice. a** Timeline indicating stereotaxic injection of the Cre-dependent AAV virus in DRN, optogenetic activation of DRN 5-HT neurons, in vivo EEG, and single-neuron recording in the prefrontal cortex and behavior tests in mice. **b** Representing immunofluorescence images (from eight independent experiments) showing co-localization of hChR2-mCherry and anti-5-HT. **c** Pulse stimulation at 10 Hz or 20 Hz as well as persistent stimulation for 120 s increased firing frequency in the same DRN 5-HT neuron in Slc6a4-hChR2 mice (left). Mean data for spike frequency in control and optogenetic stimulation groups (right). **d–f** Representative EEG traces, heatmaps of EEG frequency distribution, locomotor speed, and EEG power spectra density plots (top to bottom) before, during and after persistent optogenetic activation of DRN 5-HT neurons in Slc6a4-hChR2 mice without (**d**) or with intraperitoneal injection of 5-HT7 receptor antagonist SB-269970 (**e**) and in Slc6a4-mCherry mice (**f**). N = 8 mice. **g–i** Enlarged EEG traces (upper) of the three dashed line regions marked in d are shown together

with simultaneous single-unit recordings from three glutamatergic neurons (middle) before (**g**), during (**h**) and after (**i**) stimulation of DRN 5-HT neurons in Slc6a4-hChR2 mice. Raster plots indicate the spikes generated in these neurons. **j–l** Response thresholds to sound (**j**), electric shock (**k**), and TMT (**l**) stimulation 10 min before (black) and 5 min after optogenetic activation (pink) of DRN 5-HT neurons in Slc6a4-hChR2 mice without or with intraventricular injection of SB-269970 or in Slc6a4-mCherry mice (left). Mean data of the changed response thresholds to aversive stimuli over a 30 min period after optogenetic activation (right). **m–o** Quantification of the stay duration in the shelter after the first entry (left) or entire process (right) in response to sound (**m**), electric shock (**n**), and TMT (**o**) after persistent optogenetic stimulation in Slc6a4-hChR2 mice without or with intraventricular injection of SB-269970 and in Slc6a4-mCherry mice. All data are presented as mean ± SEM. *$P < 0.05$, **$P < 0.01$, ***$P < 0.001$; ns denoting no significant difference. For detailed statistics, see Supplementary Table 2. Source data are provided as a Source Data file.

## Methods

### Fish methods and data analysis

**Zebrafish husbandry and generation of mutant lines.** Zebrafish (Danio rerio) were maintained on a 14/10 h light/dark cycle at 28 °C in a recirculating aquatics habitats facility of Tongji University. All experiments were performed using 4–5 weeks old fish with mixed gender. Wild-type and transgenic lines used in this study included *Tg(tph2:GFP)*, abbreviated as *tph2:GFP*; *Tg(HuC:H2B-GCaMP6f)*[56], abbreviated as *HuC:H2B-GCaMP6f*; *Tg(tph2:Gal4)*[25]; *Tg(UAS:GCaMP6f)*[57]; *Tg(UAS:nfsB-mCherry)*[58]; *Tg(UAS:mCherry)*; *Tg(vglut2a:loxP-DsRed-loxP-Gal4)*, abbreviated as *vglut2a:DsRed*; and *TgBAC(gad1b:EGFP)*[59], abbreviated as *gad1b:EGFP*. All procedures were performed in accordance with guidelines of the Animal Use Committee of Tongji University.

To generate the *Tg(tph2:Kaede)*, abbreviated as *tph2:Kaede*, and *Tg(5 × UAS:ChR2(H134R)-mCherry)* lines, the pzTol2-tph2-Kaede construct containing a 3408 bp fragment from the tryptophan hydroxylase 2 (tph2) promoter[25] and the photoconvertible fluorescent protein Kaede, and the pzTol2-5 × UAS-ChR2(H134R)-mCherry construct were custom designed and produced at the Vector Builder platform. The plasmid DNA (40 ng/μL) was co-injected with Tol2 transferase mRNA (100 ng/μL) into 1 cell-stage wild-type zebrafish embryos.

The mutant zebrafish *tph2*[−/−], *htr7a*[−/−], *htr1aa*[−/−], *htr2ab*[−/−] and *htr4*[−/−] were generated using CRISPR/Cas9 technology. sgRNA was synthesized using the TranscriptAid T7 High Yield Transcription Kit (Thermo Scientific). Cas9 plasmid was digested with XbaI (Thermo Scientific) and capped Cas9 mRNA was synthesized using the mMES-SAGE mMACHINE T7 Transcription Kit (Thermo Scientific). 300 pg Cas9 mRNA together with 30 pg sgRNA were injected into embryos at the one-cell stage. Primers used for amplification of sgRNA templates and genotyping are listed in Supplementary Table 1. The *htr7c*[−/−] mutant line was generated in our previous study[60].

**Preparation of conspecific alarm substance.** One adult fish was placed in a petri dish kept on ice and conspecific alarm substance (CAS) was obtained through 10–15 superficial shallow cuts on epidermal cells with a razor blade[61]. CAS stock solution was prepared by washing on both sides of a single fish with 10 mL distilled water. To reliably induce vigilance behavior in zebrafish, 3.5 mL CAS stock solution was diluted in 1 L water.

**Conspecific alarm substance stimulation and behavior analysis.** Zebrafish of wild-type or mutant lines was placed individually in an observation tank filled with 1.5 L water, which was illuminated with a 13 W white LED light plate. After 10 min of familiarization, a 30-min video recording was performed through the side of the tank using a digital camera (25fps, MER-U3-L, DaHeng Image). The CAS or water as a control substance was dropped directly onto the water surface in the center of the tanks. For fish in the new tank experiment, they were

transferred to a new tank or dish with fresh water at 30 s after CAS treatment. The video recordings were subjected to swimming tracking and locomotor speed analysis.

**Aversive stimulation and response threshold tests.** To evaluate the response thresholds to aversive stimuli during the vigilance behavior state, we used sound, electric shock, HCL, and cadaverine (a major product of zebrafish tissue decay which can elicit olfactory-mediated avoidance response in zebrafish[62]) induced C-startle response or avoidance. The response threshold is measured as the tail angel when C-startle response or avoidance is successfully induced by the above stimuli. In each trial, individual fish was allowed to freely swim in a 6 cm petri dish, and video recording was performed from above the dish using a digital camera (10fps, EoSens CL, MIKROTRON). Video acquisition was performed with DVR Express Core (IO Industries) and controlled by CoreView software v2.1.0.33. The response threshold was tested by administration of each stimulus with increasing intensity gradient at 10 min before and every 5 min after CAS or water stimulation for a period of 30 min. For the sound respond test, sound stimulation was controlled through a DDS based Arbitrary Waveform Generator (FY6800, FeelTech) connected to an amplifier and transmitted to the speaker. The sound response threshold was tested by adjusting the amplitude from 0 V (60 dB), with stepwise increments of 1 V at 10-s intervals. The response threshold for electric shock was measured by administrating a single pulse of 50 ms with an initial intensity of 0.1 mA and then stepwise increments of 0.1 mA at 30-s intervals. Response thresholds for HCl and cadaverine were tested with concentrations from 0.05 mM to 1 mM and 1 μM to 20 μM, respectively. Fish was transferred to fresh water before testing with a new concentration. For "new dish" experiments, a fish was transferred to a new dish with fresh water at 40 s after CAS or water treatment.

**Chemogenetic ablation.** For chemogenetic ablation of 5-HT neurons, *Tg(tph2:Gal4)* was crossed with *Tg(UAS:nfsB-mCherry)* or *Tg(UAS:mCherry)* to generate *Tg(tph2:Gal4;UAS:nfsB-mCherry)*, abbreviated as *tph2:nfsB-mCherry*, and *Tg(tph2:Gal4;UAS:mCherry)*, abbreviated as *tph2:mCherry*, respectively. For the two-photon calcium imaging analysis, they were further crossed with *HuC:H2B-GCaMP6f*. For cell ablation, 4-week-old fish were treated with 5 mM metronidazole (MTZ, Sigma–Aldrich) for five days (10 h per day). During the MTZ treatment, zebrafish were maintained in the darkness to avoid photodegradation. To preclude a non-specific effect of MTZ treatment, Tg(tph2:Gal4;UAS:mCherry) fish were used as control. Neuronal activity of the Dc region and locomotor speed in response to CAS treatment were subsequently measured.

**Two-photon calcium imaging and synchrony correlation analysis.** After being anesthetized in the extracellular solution (134 mM NaCl, 2.9 mM KCl, 2.1 mM CaCl2, 1.2 mM MgCl2, 10 mM HEPES, and 10 mM

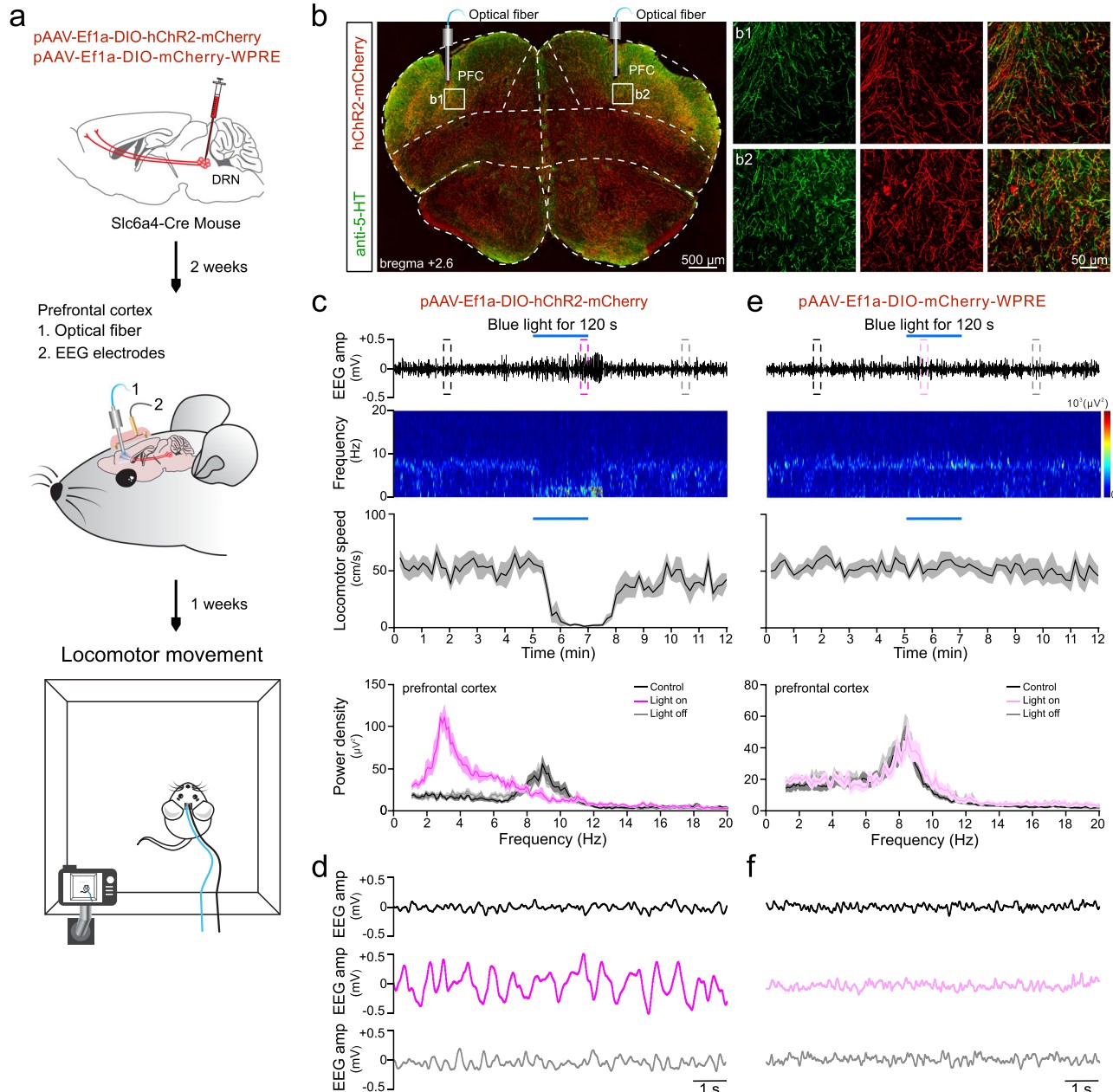

**Fig. 7 | Persistent optogenetic activation of DRN 5-HT neuron terminals in PFC induce internal synchronized state and vigilance behavior in mice. a** Drawing shows stereotaxic injection of the Cre-dependent AAV virus into the DRN, optogenetic activation of DRN 5-HT neuron terminals, in vivo EEG recording in the prefrontal cortex, and the locomotion test in mice with a timeline.
**b** Immunofluorescence image showing co-localization of hChR2-mCherry and anti-5-HT staining of DRN 5-HT neuron terminals in mice PFC. Image represents results from eight independent experiments. **c–f** Representative EEG trace, heat map of EEG frequency distribution, analysis of locomotor speed, and EEG power spectra density plot (top to bottom) before, during, and after 120 s optogenetic activation of DRN 5-HT neuron terminals in Slc6a4-hChR2 mice (**c**) and in Slc6a4-mCherry mice (**e**). The dashed line rectangles outlined in (**c**) and (**e**) are shown enlarged in (**d**) and in (**f**), respectively. $N = 8$ mice in each group.

glucose, pH 7.8, 290 mOsm) containing 0.03% MS-222, the zebrafish was fixed in the customized chamber in a dorsal side up position with insect pins and continuously superfused with extracellular fluid. For Dc regional imaging, the skull was carefully removed to expose the dorsal pallium region without damaging the brain tissue. For DRN 5-HT neuron imaging, the skin and tissue of the cerebellum and the hindbrain were removed using surgical scissors to expose the DRN. Neuronal activity of the Dc region after treating with CAS or 5-HT (Sigma−Aldrich) was detected in *HuC:H2B-GCaMP6f* line with pan-neuronal nucleus-localized GCaMP6f, crossed with $tph2^{-/-}$ or $htr7a^{-/-}$ or preincubated with the 5-HT7 receptor antagonist SB-269970 (Tocris).

5-HT was dissolved in saline at a working concentration of 10 μM and SB-269970 was dissolved in DMSO at a working concentration of 0.3 μM. To detect neuronal activity in the Dc after chemogenetic ablation of DRN 5-HT neurons, *HuC:H2B-GCaMP6f* was crossed with *tph2:nfsB-mCherry*. For detection of neuronal activity of DRN 5-HT neurons in response to CAS stimulation, *Tg(tph2:Gal4)* was crossed with *Tg(UAS:GCaMP6f)*, abbreviated as *tph2:GCaMP6f*. Calcium imaging was performed by resonant scanning with a two-photon (900 nm) laser scanning microscope. Six planes of images were captured using raster scanning pattern with a 16× water-immersion objective (N. A., 0.80; Olympus). Frames were acquired using SciScan 1.3 software

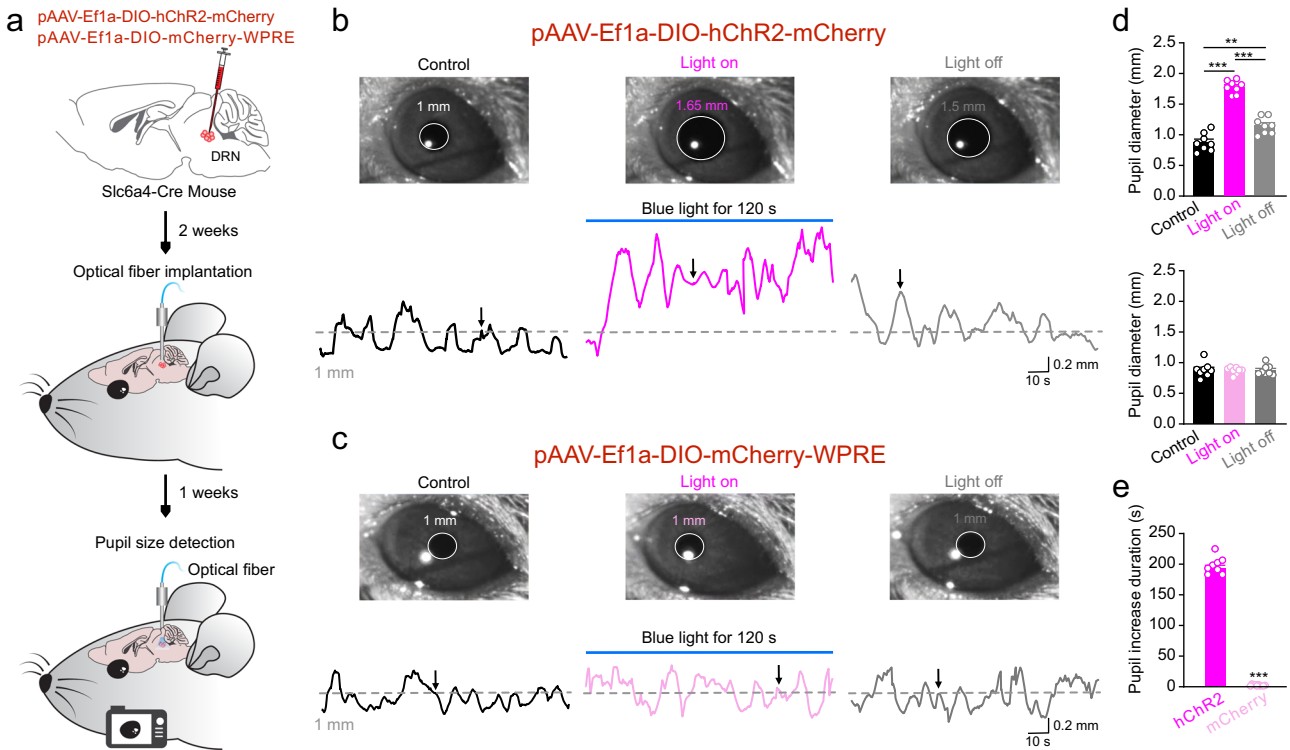

**Fig. 8 | Persistent optogenetic activation of DRN 5-HT neurons continuously increased pupillary size in mice. a** Illustration of pupillometry and optogenetic stimulation of DRN 5-HT neurons. **b**, **c** Representative pupil images (indicated by arrow) and pupillary size fluctuations in Slc6a4-hChR2 (**b**) and Slc6a4-mCherry (**c**) mice before, during, and after 120 s optogenetic activation of DRN 5-HT neurons. **d** Mean data quantifying the increase in pupillary size in Slc6a4-hChR2 and Slc6a4- mCherry mice before, during, and after optogenetic stimulation. **e** Mean data quantifying the duration of the pupillary increase in Slc6a4-hChR2 and Slc6a4-mCherry mice. $N = 8$ mice in each group. All data are presented as mean ± SEM. **$P < 0.01$, ***$P < 0.001$. For detailed statistics, see Supplementary Table 2. Source data are provided as a Source Data file.

(Scientifica), with the acquisition rate at 2.58 frame/s and image size of 1024 × 1024 pixels. Neural dynamics of subregions was analyzed in volumetric images of six layers covering the dorsal pallium. Region of interests (ROIs) indicating each single neuron in the Dc region were drawn manually for calcium signal analysis.

Correlation analysis of neuronal activity of Dc neurons was performed as previously described[63]. Firstly, the change in calcium signal was measured as the sum of pixel responses within each ROI as a function of mean fluorescence intensity ΔF/F, and the traces of calcium activity dynamics for each single neuron were calculated and smoothed with span at 0.01 using Matlab (Mathworks) scripts[51]. By thresholding at 0.5 ΔF/F (to remove noise), individual events were identified within the ΔF/F traces for each neuron in each frame when the activity signal increased above the threshold value. The time points of each event were calculated using Spike2 7.2 (Cambridge Electronic Design). To calculate the synchrony across the Dc neurons, pairwise Pearson correlation analysis was performed.

**Electrophysiology.** Animal models were prepared in the same way as for the two-photon calcium imaging. For extracellular recordings of DRN 5-HT neurons *tph2:GFP* 4-week old fish were used, in which target neurons were visualized using a fluorescent microscope (BX50WI, Olympus) and an infrared CCD camera (IR1000, DAGE-MTI). The recording electrode was filled with extracellular solution, and advanced into the brain through a motorized micromanipulator (Scientifica).

Whole-cell recordings of synchronized or non-synchronized Dc neurons in response to CAS stimulation was performed in the *HuC:H2B-GCaMP6f* fish with a two-photon microscope (Scientifica). Whole-cell recordings of excitatory and inhibitory neurons in the Dc region after treating with CAS or 5-HT were performed in *vglut2a:DsRed* and *gad1b:EGFP* lines, respectively, as well as in *vglut2a:DsRed* crossed with *htr7a$^{-/-}$* or preincubated with a 5-HT7 receptor antagonist or a SK channel activator NS309 (Tocris). NS309 was dissolved in saline at a working concentration of 100 μM. For whole-cell recordings of Dc neurons, a small incision was made in the skin above Dc and an electrode filled with intracellular solution (120 mM k-gluconate, 5 mM KCl, 10 mM HEPES, 4 mM Mg$_2$ATP, 0.3 mM Na$_4$GTP, 10 mM Na-phosphocreatine, pH 7.4 adjusted with KOH, 275 mOsm) was advanced into the Dc and approached the target neuron while applying constant positive pressure through a motorized micromanipulator (Scientifica). Signals were recorded in current-clamp gap-free mode, detected by a MultiClamp 700B amplifier (Molecular Devices) and a 1550B Digidata data acquisition system (Molecular devices) with the low-pass filtered at 10 kHz.

**Photo-conversion of Kaede.** Photo-conversion of Kaede was performed with a confocal microscope (FV3000, Olympus) equipped with a LSM module and 405 nm laser. A zebrafish (4-weeks old) was anesthetized and mounted on the chamber in extracellular solution (mentioned above) with the skull removed. The Dc region was identified. Kaede-expressing 5-HT axons at Dc region were continuously exposed to 40% of the maximum light output of the 405 nm laser for 30−60 s with a 40× water-immersion objective (Olympus). Confocal image stacks of the brain region containing both dorsal pallium and DRN were collected immediately after laser exposure and 6 h later after the converted Kaede diffused into the 5-HT soma.

**Immunohistochemistry.** A 4-week-old zebrafish was placed in chilled extracellular fluid and anesthetized. The brain tissue was dissected out

and fixed in 4% PFA for 24 h. Then the tissue was washed three times each for 5 min with PBS and permeabilized in PBS containing 1% Triton X-100 (PBST) for 2 h. After 1 h of blocking with 5% BSA in PBST, the brain tissue was then incubated in anti-serotonin (Sigma–Aldrich S5545, 1:3000) for 48 h at 4 °C. After washing the primary antibody with PBS, the tissue was incubated with secondary antibody Alexa Fluor 488 Donkey anti-Rabbit (Jackson Labs 711-545-152, 1:200) overnight at 4 °C. After washing the secondary antibody with PBS, the brain tissue was placed on a slide with the dorsal side up and mounted with the anti-fading fluorescent mounting medium (Vectorlabs) for imaging. An Olympus FV3000 laser scanning confocal microscope was used for image acquisition.

**Quantitative RT-PCR.** QRT-PCR was performed to measure the relative expression level of *htr1aa*, *htr2ab*, *htr4*, and *htr7a* in mutant lines and corresponding wild-type siblings (Supplementary Fig. 11). It was conducted on LightCycler 96 system (Roche) using SYBR Green Realtime PCR Master Mix (TOYOBO) following the manufacture's protocols. Relative expression was normalized to *gapdh*. PCR primers were listed in Supplementary Table 1.

**LC–MS analysis.** The amount of 5-HT release from dorsal pallium was quantified by LC–MS analysis of the elution of dissected dorsal pallium from 4-week-old wild-type zebrafish. Three groups (untreated control, 5 min, and 1 h post CAS stimulation) were sampled. Triplicates were done for each group with dissected dorsal palliums from five fish mixed for each. After three washes, the dorsal pallium tissue was transferred to 0.6 mL microcentrifuge tubes containing 100 μL PBS and incubated for 30 min at 28 °C. The supernatant was collected by high-speed centrifugation at 5000 g for 5 min, and the remaining tissue weight was measured and used for standardization of 5-HT concentrations. An additional 20-fold dilution of PBS was used for standard and quality control for all samples. The 5-HT concentration was determined by LC–MS analysis carried out on a Triple Quad 6500 + LC–MS/MS System equipped with an ExionLC UHPLC unit (AB SCIEX, CA, USA) as previously described[60]. Briefly, chromatographic separation was achieved on a CORTECS HILIC Column (2.1 × 100 mm, 2.7 μm, Waters Corporation, MA, USA) at 35 °C. The mobile phase consisted of water (A) and acetonitrile (B), both containing 0.1% formic acid, at a flow rate of 0.6 ml/min. The mass spectrometric detection was performed using multiple reaction monitoring with an electrospray ionization source in positive mode. Data acquisition and processing were performed with the Analyst software 1.7.1 from AB SCIEX.

**Single-cell RNA-sequencing.** Fish were fixed in a recording chamber containing extracellular fluid, and neurons with synchronized or nonsynchronized calcium activity after CAS treatment were identified by two-photon imaging and collected separately with glass pipette. Single cells were transferred to the lysis buffer solution[64]. Total RNA was extracted using TRIzol (Invitrogen) and then purified on RNeasy columns (Qiagen). Total RNA quality was assessed on a bioanalyzer (Thermofisher). RNA-Seq libraries ($N = 10$) for each group were prepared using the Illumina TruSeq Strand mRNA Prep Kit according to the manufacturer's instructions. RNA-Seq libraries were sequenced using Illumina NextSeq 500, generating 75 bp paired-end reads for each sample. RNA-Seq reads untrimmed. Differential expression analysis was performed using the CPM (counts per million) function in the Bioconductor package edgeR (v3.14.0). Low-expression genes were excluded to make a simple correction for gene counts. Genes with $P$-value (instead of adjusted $P$-value) <0.05 were assigned as differentially expressed.

**Optogenetic stimulation.** For optogenetic activation of DRN 5-HT neurons, *Tg(tph2:Gal4)* was crossed with *Tg(5 × UAS:ChR2(H134R)-mCherry)* to generate *Tg(tph2:Gal4;UAS:ChR2(H134R)-mCherry)*,

abbreviated as *tph2:ChR2-mCherry*, and *tph2:mCherry* was used as negative control. Polygon 1000 photostimulation device (Mightex) was used to control illuminated region specifically at DRN of zebrafish using a 470 nm blue light of 15 mW for 120 s, which precisely activated the DRN 5-HT neurons. Extracellular recording of DRN 5-HT neurons was performed to evaluate the activation efficiency. For whole-cell recording of glutamatergic neurons in the Dc region, *tph2:ChR2-mCherry* was crossed with *vglut2a:DsRed*, which facilitated us to perform whole-cell recording during blue-light stimulation of DRN.

For behavioral analysis, two LED light panels (12 cm × 22 cm) with adjustable light intensity were placed above and below the tank, respectively. 470 nm blue light of 100% intensity (15 mW) and 50% intensity (7.5 mW) was used for optogenetic stimulation. Zebrafish were placed individually in the observation tank. After 10 min of adaptation, a 30-min video recording was performed on the side of the tank using a digital camera (25fps, MER-U3-L, DaHeng Image). The duration of blue-light stimulation was set for either120 s or 30 s, respectively. The video recordings were subjected to swimming tracking and locomotor speed analysis. Aversive stimulation and response threshold tests after blue-light stimulation for 120 s were performed as described above.

## Mouse methods and data analysis

**Animals.** All experiments were performed in accordance with the protocols approved by the Ethics Committee for Animal Experimentation and strictly followed the Guidelines for Animal Experimentation of Tongji University. Male Slc6a4-Cre mice of two-month-old were purchased from Shanghai Southern Model Biotechnology Co., Ltd (Shanghai, China) and used for all in vivo and ex vivo experiments. Mice were housed in the Laboratory Animal Center of Tongji University on a 12-h light-dark cycle (lights on at 8 a.m., lights off at 8 p.m.) at 20–26 °C and humidity of 40%–70% with ad libitum access to food and water with *ad libitum* access to food and water.

**Virus and drugs.** Virus of pAAV-Ef1a-DIO-hChR2(H134R)-mCherry or pAAV-Ef1a-DIO-mCherry-WPRE were used for optogenetic experiments, pAAV-hSyn-DIO-hM3d(Gq)-mCherry or pAAV-hSyn-DIO-mCherry-WPRE were used for chemogenetic experiments. All the virus tools were packaged by OBIO Technology (Shanghai, China). For pharmacological experiments, a selective 5-HT7 receptor antagonist SB-269970 (Tocris) was injected intraperitoneally at a concentration of 10 mg/kg or injected intraventricularly at a concentration of 20 μg/μL. 5-HT (Sigma–Aldrich) at a concentration of 20 μg/μL was injected intraventricularly. Clozapine N-Oxide (CNO, Sigma–Aldrich) was dissolved in saline and used at a working concentration of 2 mg/kg for intraperitoneal injection and 10 μM for electrophysiological recordings in brain slices.

**Surgical procedures.** Mice were anesthetized with isoflurane (2% for induction, 0.5%–1% for maintenance, and a flow rate of 0.8 L/min). Then the mouse was placed in the stereotaxic instrument (68001, RWD) on a heating pad with the temperature set to 37 °C. The hair above the skull was removed and the skin was sterilized with iodophor.

For AAV virus injection, 0.5 μL of virus volume was injected using a 5 μL Hamilton syringe. The injection needle was placed into the DRN (bregma: AP = −4.4 mm; ML = 0 mm; DV = −3 mm) and withdrawn 10 min after injection.

For optogenetic manipulation of DRN 5-HT neurons, a 200 μm optical fiber (Inper) was slowly advanced into the DRN area until the tip was 100 μm above the virus injection site. Then the optical fiber was fixed in place in the skull with dental cement. A similar procedure was followed for optogenetic manipulation of DRN 5-HT neuron axon terminals in the PFC, except that here the optical fiber was implanted into the PFC area (bregma: AP = +2.6 mm; ML = ±1 mm; DV = −1.5 mm).

For optogenetic manipulation in pupillometry experiments, a screw of 3 mm diameter and 1.5 cm long was mounted into the skull surface at lambda of the mouse, which was then used as a head-fixation holder after implantation of the optical fiber.

For optogenetic manipulation in single-unit and EEG recording, a 32-channel array was implanted into the mouse prefrontal cortex, after implantation of the optical fiber. Two holes were drilled around the skull and cranial nails were embedded for the connection of EEG electrodes (frontal region, bregma: AP = +1.50 mm; ML = −1.50 mm; DV = −1.5 mm) and reference ground electrode (bregma: AP = −3.0 mm; ML = +2.0 mm; DV = −1.5 mm). In addition, a square window of 1 mm$^2$ was ground into the upper layer of the skull over the target brain area, and subsequently multi-channel electrodes were embedded into the prefrontal cortex (bregma: AP = +2.6 mm; ML = ±1 mm; DV = −1.5 mm) with a decline rate at 50 μm/min. Medical brain glue (World precision instruments) was used to fix electrode and fill the exposed space of the skull. Then the entire electrode system was fixed with dental cement.

For intracerebral cannula plantation, after carefully exposing the skull, two small holes (AP = +2.6 mm; ML = ±1 mm; DV = 1.5 mm) were drilled in the skull over the prefrontal cortex on both sides of the cerebral hemispheres to facilitate cannula insertion. The guide cannula (RWD) was placed through the drill hole over the prefrontal cortex and subsequently fixed to the skull surface with dental cement. On the day of behavioral tests, the internal stylet was removed from the guide cannula, a stainless-steel infusion cannula was inserted and extended 0.5 mm below the tip of the guide cannula. The drug (total volume 1 μL) was infused at a rate of 0.5 μL/min through a polyethylene tube connected to a microsyringe. The infusion cannula was removed 5 min after cessation of infusion and the internal test needle was reinserted.

For chemogenetic experiments, CNO or vehicle was injected intraperitoneally 40 min before behavior tests. Each behavioral paradigm was separated by a wash-out period of at least 1 week.

After virus injection, mice were left for 3 weeks to ensure sufficient viral transgene expression. Surgical procedures were performed at 2 weeks post virus injection, and mice were returned to their home cages for individual care and allowed to recover for 1 week before behavioral tests and electrophysiological experiments.

**Optogenetic stimulation.** Blue light (465 nm) was generated with a B2-465 laser (Inper) and delivered to the DRN region or PFC. For persistent activation of DRN 5-HT neurons, blue-light stimulation of 5 mw, 10 mw or 15 mw was applied for 120 s or 300 s. Persistent light stimulation of 15 mW for 120 s was applied since it induced more stable vigilance behavior and higher EEG power density than other stimulation modes. For pulse stimulation, 15 mW blue-light at 10 Hz or 20 Hz for 120 s or 300 s were used. For optogenetic stimulation of DRN 5-HT axon terminal in PFC, 15 mW for 120 s was used.

**In vivo EEG and single-unit recording and analysis.** To monitor the activity dynamics of PFC neurons after optogenetic activation of DRN 5-HT neurons or axon terminals in PFC, 12 min of activity was recorded consisting of 5 min pre, 2 min during, and 5 min post-light activation. Broadband (0.3 Hz–7.5 kHz) neural signals (16 bits at 30 kHz) were recorded in real-time using a data acquisition system (Zeus, Bio-Signal Technologies) via an implanted 32-channel array (15 microfilament electrodes per half brain, two additional channels connected to the EEG screw and the reference ground screw, respectively). Spikes were detected with a high pass (300 Hz) filter. Real-time spike classification was conducted using principal component analysis (PCA). Spike categorization was optimized using Offline Sorter ×64 V4 (Plexon) before analyzing the data in NeuroExplorer 5 (Nex Technologies). For neuron type classification, the peak-trough of the waveform was calculated and the unit was classified as a glutamatergic broad spiking cell if the mean width was greater than 400 μs, otherwise it is a GABAergic

narrow spiking cell. The EEG signal was filtered by Band-Pass Filter from 1 to 40 Hz. The EEG signal power density and spectrograms were computed by NeuroExplorer 5. The frequency bands were composed of delta (δ: 1–4 Hz), theta (θ: 4–8 Hz), alpha (α: 8–13 Hz), beta (β: 13–30 Hz), and gamma (γ: 30–40 Hz) waves. Locomotor movement was simultaneously recorded via video tracking for locomotor speed analysis.

**Pupillometry.** The mouse was placed in an anterior-posterior open cylinder tube so that it could move its limbs when the head was fixed. The lens for pupil tracking was located 20 cm away parallel to the pupil, and an infrared light source was used for pupil recognition. Pupil diameter was measured with Oculomatic Pro 1.9.7. (Bio-signal Inc) at a sampling frequency of 1 kHz. During optogenetic activation, pupil size was recorded for 12 min consisting of 5 min pre, 2 min during, and 5 min post-light activation. For chemogenetic activation, pupil size was recorded for 1 h before and 40 min or 120 min after CNO injection. The traces of pupil size dynamics were generated and smoothed using a Matlab scripts[51].

### Behavioral analyses

**Open field test.** Mice were placed in an open field box (40 cm × 40 cm × 40 cm), and free movement was tracked by the Smart V3.0 behavioral analysis system (RWD) and the total distance moved was recorded. During optogenetic experiments, locomotor movement was recorded for 12 min synchronously with the in vivo single-unit recordings. For chemogenetic experiments, locomotor movement was recorded after administration of CNO.

**Response threshold test.** To detect the response thresholds during vigilance-like behavior, three aversive stimuli namely sound, 2,3,5-Trimethyl-3-thiazoline (TMT), and electrical shock were applied to mice after optogenetic or chemogenetic activation of DRN 5-HT in the open field box equipped with a camera. The response threshold for different stimuli was defined as the time taken by the mouse to escape or move away from the stimulation area. Escape was distinguished from random exploration behavior when mice moved at a speed 10% higher than in the 5 s before exposure to the stimulus. For the sound response threshold test, an amplifier (XT25G30-04 1″ Dual Ring Tweeter, Vifa) was attached to one of the walls to transmit the sound. The stimulation frequency was 2 kHz whose intensity increased by 5 dB every 15 s from a starting value of 50 dB. For the TMT response threshold test[65], 5 uL of TMT solution (pure or 1:10$^1$, 1:10$^2$, 1:10$^3$, 1:10$^4$, 1:10$^5$, 1:10$^6$, 1:10$^7$, 1:10$^8$ dilutions) was applied to a cotton ball which was placed in the corner of the observation box. Mice were tested for 1 min for each concentration. After each concentration test, the observation box was cleaned with 75% alcohol to eliminate the TMT odor. For response threshold of electrical shock, the duration of electrical stimulation was 0.5 s, with intensity starting at 0 mA and increasing by 0.02 mA every 15 s. In order to prevent habituation, only one round of stimulations was performed per day for each mouse. For optogenetic experiments, the response threshold was tested by administration of each stimulus with intensity gradient at 10 min before and every 5 min after blue-light stimulation. For chemogenetic experiments, response threshold was tested 40 min after CNO injection. For administration of 5-HT by intracerebral cannula, response threshold test was also performed till 30 min after 5-HT application.

**Shelter tests.** A shelter (6 cm × 6 cm × 6 cm) was placed in the corner of an open field box for assessing vigilance behavior after optogenetic or chemogenetic activation of DRN 5-HT neurons or administration of 5-HT by intracerebral cannula. Three aversive stimuli (sound, TMT or electricity) were chosen for shelter test. The stimulation intensities selected were 20% higher than the response threshold detected in the

experiments described above. The stay time in the shelter from first entry after treatment were used to assess vigilance state.

**Electrophysiological recording.** Mice were anesthetized with ether and rapidly decapitated. The brain was removed and placed in ice-cold artificial cerebrospinal fluid (ACSF), including 120 mM NaCl, 2.5 mM KCl, 2 mM CaCl$_2$, 2 mM MgSO$_4$, 1.2 mM NaH$_2$PO$_4$, 26 mM NaHCO$_3$, and 10 mM D-glucose. Brain slices (300 μm) were obtained using a Vibratome (VT1000s Leica) while bathed in a slice-cutting solution consisting of 220 mM sucrose, 2.5 mM KCl, 1.3 mM CaCl$_2$, 2.5 mM MgSO$_4$, 1 mM NaH$_2$PO$_4$, 26 mM NaHCO$_3$, and 10 mM D-glucose. Brain slices were equilibrated in ACSF in a holding chamber at a temperature of $33 \pm 1\,°C$ for 30 min, followed by an additional 2–8 h at room temperature ($25 \pm 1\,°C$). All solutions were saturated with 95% O$_2$/5% CO$_2$.

For electrophysiological recording, mCherry-positive DRN 5-HT neurons were visualized using a fluorescent microscope and infrared CCD camera. For optogenetic activation, 5-HT neurons expressing ChR2 were stimulated with a blue-light (470 nm) LED using either a persistent or a pulse stimulation program. For chemogenetic activation, CNO (10 μM) was added to the superfusate. The patch electrode was filled with the following solution containing 105 mM K-gluconate, 30 mM KCl, 10 mM HEPES, 10 mM phosphocreatine, 4 mM Mg$_2$ATP, 0.3 mM Na$_4$GTP, 0.3 mM EGTA (pH 7.35, 285 mOsm). The spontaneous activity of neurons was recorded under current-clamp mode.

**Immunohistochemistry.** To verify viral expression, mice were deeply anaesthetized with isoflurane and transcardially perfused with PBS, followed by fixation in cold 4% PFA solution for 48 h. Then the brain tissue was dissected and fixed in PFA overnight at 4 °C. The brain tissue was dehydrated in PBS solution containing 30% sucrose and then embedded in OCT freezing embedding solution. Coronal sections (30 μm) were cut on a Leica CM1860 UV freezing microtome (20 sections per brain). The brain slices were permeabilized in PBST for 20 min, subsequently blocked with 5% BSA in PBST for 1 h at room temperature, and then incubated with anti-mCherry (Thermo Fisher M11217,1:1000) and anti-Serotonin (Sigma–Aldrich s5545,1:2000) overnight at 4 °C. After washing with PBS for 10 min and PBST twice for 10 min each, brain slices were incubated in Alexa Fluor 568 Goat anti-Rat (Thermo Fisher, 1:400) and Alexa Fluor 488 Donkey anti-Rabbit (Jackson Labs, 1:200) for 3 h at room temperature. After washing 3 times with PBS for 10 min each, brain slices were sealed with anti-fading fluorescent mounting medium (Vectorlabs) overlay and subsequently imaged with a confocal microscope.

**Data analysis and statistics.** Quantification of the number of neurons on confocal images was performed using Fiji v1.53c (NIH), and images were edited with Photoshop CC 2018 (Adobe) and Fiji. Videos were edited with Premiere Pro CC 2018 (Adobe). Zebrafish behavior tracking and locomotor speed statistics was conducted using Python-based self-contained ZebraZoom v1.17 software[66]. Clampex v10.7.0.3 and Clampfit v10.6 (Molecular Devices) software was used for collection and analysis of electrophysiological signals. Matlab R2018a v9.4 scripts were used for generating calcium signal curves and heatmaps. Smart V3.0 behavioral analysis system was used for behavior tracking and locomotor speed statistics in mice. Oculomatic Pro 1.9.7 system was used for pupil size detection in mice. NeuroExplorer 5 software was used for EEG spectrum production and statistics. Data were presented as mean ± S.E.M. Statistical difference was assessed using one-way ANOVA or students' unpaired or paired two-tailed $t$-tests using Prism 9.0 (GraphPad software), and $p < 0.05$ was considered statistically significant. The experimenter was blind to the condition of each animal when analyses were conducted.

**Reporting summary**

Further information on research design is available in the Nature Portfolio Reporting Summary linked to this article.

## Data availability

Raw and processed RNA-seq data generated in this study are deposited into the GEO database with accession number GSE253039. Source data are provided with this paper.

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

## Acknowledgements

We thank Drs. M Luo, H Li, and R.F. Oliveira for comments on the study; Drs. A El Manira, H.A. Burgess, J-L Du, and China Zebrafish Resource Center (CZRC) for sharing the fish lines; J Mao for fish line generation and care. This work was supported by STI2030-Major Projects (2021ZD0204500 and 2021ZD0204501), the National Natural Science Foundation of China (32320103004, 31972904, 32271051), Shanghai Municipal Science and Technology Major Project (2018SHZDZX05) and Shanghai Blue Cross Brain Hospital Co., Ltd.

## Author contributions

Y.Z. did most experiments with help from C-X.H. and Y.G., Y.Zhao., W.R., Y.W., and J.C. helped generate the transgenic lines. Y.Z., C-X.H. and N.N.G. analyzed data. J.S. designed the experiments, interpreted the results, and wrote the manuscript.

## Competing interests

The authors declare no competing interests.
