## [Peer Review File · Nature Communications]

Serotonergic modulation of vigilance states in zebrafish and miceREVIEWER COMMENTS

Reviewer #1 (Remarks to the Author):

In this manuscript, the authors show the raphe-dorsal pallium pathway plays a critical role in generating and maintaining vigilance state, and this novel function is conserved cross fish and mammal. The authors first found that exposure to CAS can reliably trigger vigilance state on adult zebrafish, at which fish's response thresholds to multiple types of aversive stimuli significantly decrease. By imaging population activity of serotonergic neurons in DRN and dorsal pallium neurons, the authors revealed Dc region synchronization is highly correlated with the generation and maintenance of vigilance state, and the synchronization is achieved through DRN-Dc projection of DRN serotonergic neurons. In addition, combining whole-cell recording and Smart-seq2 analysis, the synchronized glutamatergic neuron in Dc region was identified, whose activity could be inhibited by DRN 5-HT neurons via 5-HT7 receptors. Furthermore, the authors causally validated this pathway in both fish and mouse models, indicating the raphe-dorsal pallium pathway may play a conserved role cross phyla.

Overall, the experiments are well designed and performed, and the study is well organized and provides novel insights on ancestral neural mechanism of how behavioral states are triggered and maintained at neural circuit level. A few of issues need to be clarified to further strengthen the whole story.

Major concerns:

1. As the pretectum and pineal gland in fish also have 5-HT neurons, it is necessary to exclude their involvement in vigilance state regulation.
2. In Figures 2F, 3P and 3Q, as well as in S2, the results showed most of post-CAS activated neurons are in anterior and medial dorsal pallium, whereas in Figures 3J and 3K, the DRN 5-HT fibers in anterior and medial dorsal pallium are on relatively sparse. It would be helpful to present morphological evidence to show fibers of Dc-projecting DRN 5-HT neurons and Dc glutamatergic neurons have contacts, or at least, their fibers may co-localize.
3. Concerning the raphe-dorsal pallium circuit mechanism, following issues need to be clarified.
 - 3.1 How long could the DRN-5HT neurons keep their high frequency firing pattern?
 - 3.2 There seems to be a mismatch in DRN firing rate increase and the Dc synchronization. The firing frequency of DRN 5-HT neurons significantly increase after CAS and keep this high-frequency firing for at least one minute (Figure 3G), whereas apparently only very few of these spikes induced synchronization in Dc (in Figure 2D and Figure 3, the interval of synchronized oscillation is about 120 s). Why the DRN-5HT neurons are constantly activated for more than 100 s according to Figure 3C, whereas the synchronization induced by this pathway happens only about one time in 120 s according to Figure 2D? Is there a threshold for triggering Dc synchronization?
 - 3.3 In Figure 3C, activation level of DRN 5-HT neurons varies. Does the latency of vigilance state generated (after aversive behavior) correlates with the activation intensity or numbers of DRN 5-HT neurons?
4. It would be nice to present direct evidence showing temporal relations of DR activation, Dc synchronization and aversive or vigilance behavioral response after CAS. The authors might record EMG while monitoring DRN 5-HT and Dc region activity to give a better sequential description.

Minor concerns:

1. In Figure 4C, it looks the calcium oscillation trace is so silent when the synchronized neurons are on burst firing. The authors shall state more clearly how the calcium oscillation traces processed.
2. About the generation of vigilance state, are aversive stimuli in other sensory modalities strong enough to trigger vigilance state of zebrafish?
3. What kind of sensory stimuli might trigger vigilance state on mice?
4. For calcium images in Figures 2, 3 and 4, color bars should be provided.

Reviewer #2 (Remarks to the Author):

Zhao et al addresses the neural circuitry underlying the induction of a state of vigilance (increased attention, reduced or abolished locomotion, lowered threshold for external stimuli like sound) in both zebrafish and mouse. It is shown that an activation of 5-HT neurons projecting to an area Dc within the dorsal pallium (cortex) in zebrafish leads to an induction of vigilance. This coincides with a transformation of the firing pattern of glutamatergic Dc neurons from regular spiking neurons to burst firing, and a slow synchronous firing of this group of Dc neurons. This effect is mediated via 5-HT₇ receptors that induce burst-firing through a depression of SK channels. So far, the data provides a correlation, but they then show optogenetically that an activation of 5-HT terminals in Dc pallial area also induces vigilance. Thus, the link between 5-HT neurons that via 5-HT₇ receptors transform pallial neurons to burst firing induce the vigilance state. In mouse they establish parallel findings with activation of 5-HT neurons in Nucleus Raphe Magnus projecting to the prefrontal cortex and there induce burst firing via 5-HT₇ receptors and vigilance behavior, and that local activation of 5-HT terminals in the prefrontal lobe can induce vigilance. In other words, they have shown that the induction of vigilance, an important survival mechanism, is elicited via a conserved 5-HT- system acting via 5-HT₇ receptors inducing similar postsynaptic effects on pallial/cortical neurons in both zebrafish and mouse. The study is thus impressive and very extensive and complete utilizing all relevant techniques (genetics, optogenetics, electrophysiology, calcium imaging, pharmacology and detailed behavioral techniques) to establish these findings. It is well illustrated with 8 detailed figures, 11 supplementary and 5 videos. The text reads well. The Result section and Methods are well-structured, but the Discussion section could be shortened somewhat and be structured in a more succinct way.

I have mainly minor suggestions.

Comments,

In the discussion, it may be worthwhile discussing briefly the possible projection pattern of the pallial/cortical neurons shown to mediate the vigilance behavior. They could involve both other telencephalic structures and subcortical structures like PAG and hypothalamus. To unravel this will of course represent a future detailed study.

Specific comments

Line 36 define Dc region, as pallial DC region.

Line 51, suggest...cortex or other parts of CNS...

Line 55, suggest...or dangers and avoid detection by not moving.

Lines 88-89 suggest: "...5-HT₇ receptors. Genetic ablation or pharmacological blockade of 5-HT-7 receptors abolished the vigilance behavior".

Line 121, should be hemispheres.

Line 142 suggest ..discharging at a much higher frequency in... Comment .a discharge a frequency of around 3Hz is not "high frequency".

Line156 Help the reader by defining what tph stands for.

Line 161-164. Too brief description of the results.

Lines 226-229 same as line 156

Line 294 should be ...simultaneous...

Line 340-341 suggest...in the pallial DC-region of zebrafish or in the prefrontal cortex...

Line 346 suggest. ...spontaneous single spike firing to.

Line 349 suggest: ...the neuronal populations,

Line 353 suggest:...alert..

Line 385 suggest exchange extraordinarily to much.

Line 1142 Define, "cadaverine" and difference to CAS

Line 1145 Suggest: ""All data here and further below will be presented as...""

Figure 2G. The interpretation of the cross-correlation squared diagrams, control and CAS and the color code need further explanation, also since they are used on other below.

Figure 5A expand explanation.

Figure 6E, simplify fro the reader by labelling with 5-HT7 antagonist.

Supplementary figures:

All supplementary figures should have a title of the legend, similar to the main figures and e.g. Figure S7. Perhaps some more detail in these legends.

Figure S3C. Explain the procedure better and the effect of MTZ.

The videos are excellent. However, the video legends could be expanded to include more detail. There is for instance a very long period before eg CAS is entered in video 1, and an impatient viewer may give up for the striking effect occurs.

Reviewer #3 (Remarks to the Author):

The manuscript by Zhao et al uncover a conserved link between 5-HT neurons in the dorsal raphe nuclei and cortical neurons (or their homolog) underlying vigilance in zebrafish and mouse. The authors use a combination of behavioural testing, calcium imaging, electrophysiology, optogenetics and single-cell sequencing to identify a role of 5-HT in synchronizing glutamatergic neurons in the cortex through activation of 5-HT7a. The experiments are technically challenging and the results in support of their hypothesis is strong. The findings further support the role of dorsal raphe nuclei neurons in vigilance and identify the circuitry and circuit activity that lead to a vigilant state in vertebrates.

I only have minor comments:

1. Line 136. Serotonin is often used to induce locomotor activity in the isolated spinal cord. How does this observation reconcile with the reduction of locomotor activity due to DRN 5-HT neuron activity?

2. Fig. 3R. I assume exogeneous 5-HT administration in the *tph2*^{-/-} line was in the absence of CAS. Is the same response seen in *tph2*^{+/+} line to exogeneous 5-HT (in the absence of CAS)?

3. Fig. 4C. The post-inhibitory rebound in synchronized neurons occurred at the onset of calcium oscillations yet the firing right after the PIR isn't that much higher in intensity in these neurons than in intervals between the calcium oscillations. Can you comment on how the firing after the PIR could induce calcium oscillations and synchronization but not the firing of these neurons in between calcium oscillations?

4. Lines 459-460 How was the working concentration of CAS established?

The point-to-point response to the reviewers

Reviewer #1 (Remarks to the Author):

In this manuscript, the authors show the raphe-dorsal pallium pathway plays a critical role in generating and maintaining vigilance state, and this novel function is conserved cross fish and mammal. The authors first found that exposure to CAS can reliably trigger vigilance state on adult zebrafish, at which fish's response thresholds to multiple types of aversive stimuli significantly decrease. By imaging population activity of serotonergic neurons in DRN and dorsal pallium neurons, the authors revealed Dc region synchronization is highly correlated with the generation and maintenance of vigilance state, and the synchronization is achieved through DRN-Dc projection of DRN serotonergic neurons. In addition, combining whole-cell recording and Smart-seq2 analysis, the synchronized glutamatergic neuron in Dc region was identified, whose activity could be inhibited by DRN 5-HT neurons via 5-HT7 receptors. Furthermore, the authors causally validated this pathway in both fish and mouse models, indicating the raphe-dorsal pallium pathway may play a conserved role cross phyla.

Overall, the experiments are well designed and performed, and the study is well organized and provides novel insights on ancestral neural mechanism of how behavioral states are triggered and maintained at neural circuit level. A few of issues need to be clarified to further strengthen the whole story.

Thank you for the comments on our manuscript. We have addressed these issues as follow.

Major concerns:

1. As the pretectum and pineal gland in fish also have 5-HT neurons, it is necessary to exclude their involvement in vigilance state regulation.

Response:

Thank you for the question. We have performed new experiments to evaluate the function of 5-HT neurons in both pretectum and pineal gland underlying vigilance state. First, we examined the calcium response to CAS stimulation in pretectal 5-HT neurons using *Tg(tph2:Gal4; UAS:GCaMP6f)*. These neurons were not responsive to CAS stimulation (as shown below).

The 5-HT neurons in pretectum displayed no response to CAS stimulation

Second, since 5-HT neurons in pineal gland cannot be visualized using *Tg(tph2:Gal4; UAS:GCaMP6f)* line, we did extracellular recordings in these individual 5-HT neurons using *Tg(tph2: GFP)*, and it also confirmed that the discharge of the pineal 5-HT neurons are not influenced by CAS stimulation.

The 5-HT neurons in pineal gland displayed no response to CAS stimulation

Thus, these two populations of 5-HT neurons are not involved in vigilance state regulation. Since we already got a large number of figures, we would like to show these data in the point-to-point response to reviewer, not put them in the manuscript. We hope the reviewer could understand this. Thank you!

2. In Figures 2F, 3P and 3Q, as well as in S2, the results showed most of post-CAS activated neurons are in anterior and medial dorsal pallium, whereas in Figures 3J and 3K, the DRN 5-HT fibers in anterior and medial dorsal pallium are on relatively sparse. It would be helpful to present morphological evidence to show fibers of Dc-projecting DRN 5-HT neurons and Dc glutamatergic neurons have contacts, or at least, their fibers may co-localize.

The co-localization between 5-HT fibers and Dc glutamatergic neurons. a. The whole image stack; b. The enlargement of the yellow box on a focal plate.

Response:

Thanks for the great question.

First, in the new experiments, we imaged the dorsal pallium showing both 5-HT fibers and glutamatergic neurons in the pallium in the *Tg(tph2: GFP)* crossed with *Tg(vglut2a:loxP-DsRed-*

loxP-Gal4) line. Consistent with the suggestion for the reviewer, the results showed that DRN 5-HT fibers displayed a close contact with Dc glutamatergic neurons in anterior and medial pallium. It is important to see the co-localization between 5-HT terminal and Dc glutamatergic neurons (Please see the figure above).

Second, the effect of 5-HT on Dc glutamatergic neurons could also be dependent on volume transmission of 5-HT functioning extrasynaptically. This is verified by the results showing that the synchronized neurons are 5-HT7R-expressing neurons. Thus, one explanation could be the 5-HT released by the intense innervation of 5-HT terminal diffusing to the 5-HT7R-expressing neurons and playing the role by activation of 5-HTR7.

3. Concerning the raphe-dorsal pallium circuit mechanism, following issues need to be clarified.

3.1 How long could the DRN-5HT neurons keep their high frequency firing pattern?

Response:

Thanks for the question. The calcium image data (Figure 2b-d) indicated that the timing of the peak amplitude was about 120s, which suggests that the activation time will be over 120s. In general, the most DRN 5-HT neurons can keep their high-frequency firing pattern for more than 200s as shown by extracellular recording in the data below (part of the data shown in Supplementary Figure 3). By the way, the activation time of individual 5-HT neurons seems a little diverse.

The extracellular recording of 5-HT neurons before and after application of CAS. The raster plot showing the firing frequency of 5-HT neurons across the time line.

3.2 There seems to be a mismatch in DRN firing rate increase and the Dc synchronization. The firing frequency of DRN 5-HT neurons significantly increase after CAS and keep this high-frequency firing for at least one minute (Figure 3G), whereas apparently only very few of these spikes induced synchronization in Dc (in Figure 2D and Figure 3, the interval of synchronized oscillation is about 120 s). Why the DRN-5HT neurons are constantly activated for more than 100 s according to Figure 3C, whereas the synchronization induced by this pathway happens only

about one time in 120 s according to Figure 2D? Is there a threshold for triggering Dc synchronization?

Response:

Thanks for the great question and suggestion. Yes, we think there is a threshold of 5-HT firing frequency or 5-HT local concentration in Dc to trigger Dc synchronization. We have a few lines of evidence.

Firstly, the extracellular recording of raphe 5-HT neurons before CAS showed that there is a basic discharging rate of these neurons. After CAS stimulation, the discharging rate are intensely increased (Figure 3 and Supplementary Figure 3). The increase of the discharging rate in 5-HT neurons highly correlated with onset occurrence of Dc synchrony. Secondly, the calcium imaging also showed that after CAS stimulation, the 5-HT neurons displayed a robust activation for a long time (Figure 3). Thirdly, we measured the 5-HT concentration in Dc before and after CAS. Before CAS, we did not significantly detect the 5-HT concentration. But 5 minutes after CAS stimulation, the 5-HT concentration was detected to be strikingly high. This increase of released 5-HT was metabolized in one hour, because we could not detect it again one hour after CAS stimulation. The increase and decrease of released 5-HT concentration highly correlated with the onset and decline of neuronal synchrony in Dc region across time.

Conversely, when we knocked out the *tph2* gene or performed chemogenetic ablation of 5-HT neurons, the Dc synchronization as well as the vigilance behavior were completely abolished without the released 5-HT. This effect was also observed after genetic ablation of 5-HT7R.

We also took the suggestion from the reviewer (see response to question 4), we provide the sequential activation of EMG, DRN 5-HT neurons and neuronal synchrony in Dc region.

Thus, we totally agree with the reviewer that there is a threshold in terms of the discharge of raphe 5-HT neurons or released 5-HT concentration for triggering Dc synchronization.

3.3 In Figure 3C, activation level of DRN 5-HT neurons varies. Does the latency of vigilance state generated (after aversive behavior) correlates with the activation intensity or numbers of DRN 5-HT neurons?

Response:

Thank you for the question. Yes, we agree with the opinion.

As we showed in the answer to 3.1, the activation of DRN 5-HT neurons did vary. The calcium response of DRN 5-HT neurons is very difficult to compare, since the expression level of GCaMP6s is hard to quantified between different animal preparations. To answer this question, we reanalyzed the data for optogenetic stimulation of DRN 5-HT neurons. In this configuration, the light intensity represents the activation intensity of DRN 5-HT neurons (Figure 5 and 6). As

suggested by the reviewer, we do find that the latency to generate vigilance state negatively correlated with the activation intensity of DRN 5-HT neurons. The figure below showed that the increase of blue-light intensity decreased the latency to generated vigilance state.

The increase of blue-light intensity decreased the latency to generated vigilance state

4. It would be nice to present direct evidence showing temporal relations of DR activation, Dc synchronization and aversive or vigilance behavioral response after CAS. The authors might record EMG while monitoring DRN 5-HT and Dc region activity to give a better sequential description.

Response:

Thank you for the great question and suggestion. To better describe the sequential activation, we performed new experiments combining with the EMG recording, the extracellular recording of DRN 5-HT neurons and calcium imaging of Dc neurons. The results demonstrate that the aversive activity (EMG) occurred first, then the intense activation of DRN 5-HT neurons follows and finally the neuronal synchrony in Dc appears. We wish this result answers this question. We now put these data in the supplementary figure 3b and add sentences to describe this (Page7 Line158-160).

The sequential activation of EMG, DRN 5-HT neurons and neuronal synchrony in Dc

Minor concerns:

1. In Figure 4C, it looks the calcium oscillation trace is so silent when the synchronized neurons are on burst firing. The authors shall state more clearly how the calcium oscillation traces processed.

Response: Thanks for the question.

The traces of calcium activity dynamics for each single neuron were calculated and smoothed with span at 0.01 using Matlab (Mathworks) scripts (Xu L et al., 2021). We now add this information in the Methods (Page 20, Line 513).

2. About the generation of vigilance state, are aversive stimuli in other sensory modalities strong enough to trigger vigilance state of zebrafish?

Response: First, it was reported by previous studies that stimuli like cadaverine (Hussain et al., 2013), sound/noise (Bhandiwad et al., 2018) and HCl (Lovett-Barron et al., 2020) only induce aversive behavioral response of zebrafish but cannot trigger vigilance state. Second, in our study, sound, electric shock also only induced C-startle response or avoidance.

3. What kind of sensory stimuli might trigger vigilance state on mice?

Response: This is a very great question and we are also curious with it. We tried electric shock, looming stimulation as well as Trimethylthiazole (TMT) as the olfactory stimulation and it seems not working well. This might need a specific scenery or combinatory approaches. For this issue, we will continue exploring the possibilities in the future studies.

4. For calcium images in Figures 2, 3 and 4, color bars should be provided.

Response: We understand this question. Probably, we were not clear at this point. We use these images to show the spatial location of neuron population underlying the control and the treatment groups. Thus, they only represent the base-line response (Control) and the response of the treatment in a qualitative way. The specific amplitude of calcium fluctuation was illustrated by the traces, which quantitatively showed how the neurons responding to the CAS stimulation.

Reviewer #2 (Remarks to the Author):

Zhao et al addresses the neural circuitry underlying the induction of a state of vigilance (increased attention, reduced or abolished locomotion, lowered threshold for external stimuli like sound) in both zebrafish and mouse. It is shown that an activation of 5-HT neurons projecting to an area Dc within the dorsal pallium (cortex) in zebrafish leads to an induction of vigilance. This coincides with a transformation of the firing pattern of glutamatergic Dc neurons from regular spiking neurons to burst firing, and a slow synchronous firing of this group of Dc neurons. This effect is mediated via 5-HT7 receptors that induce burst-firing through a depression of SK channels. So far, the data provides a correlation, but they then show optogenetically that an activation of 5-HT terminals in Dc pallial area also induces vigilance. Thus, the link between 5-HT neurons that via 5-HT7 receptors transform pallial neurons to burst firing induce the vigilance state. In mouse they establish parallel findings with activation of 5-HT neurons in Nucleus Raphe Magnus projecting to the prefrontal cortex and there induce burst firing via 5-HT7 receptors and vigilance behavior, and that local activation of 5-HT terminals in the prefrontal lobe can induce vigilance. In other words, they have shown that the induction of vigilance, an important survival mechanism, is elicited via a conserved 5-HT-system acting via 5-HT7 receptors inducing similar postsynaptic effects on pallial/cortical neurons in both zebrafish and mouse. The study is thus impressive and very extensive and complete utilizing all relevant techniques (genetics, optogenetics, electrophysiology, calcium imaging, pharmacology and detailed behavioral techniques) to establish these findings. It is well illustrated with 8 detailed figures, 11 supplementary and 5 videos. The text reads well. The Result section and Methods are well-structured, but the Discussion section could be shortened somewhat and be structured in a more succinct way.

Thank you very much for the comments on our manuscript. As suggested by the reviewer, we have shortened the discussion (Page14-16, Line332-409).

I have mainly minor suggestions.

Thanks for all these suggestions and we have corrected them accordingly.

Specific comments

Line 36 define Dc region, as pallial DC region.

Response: As required by the journal, we shortened the abstract. We define it as the pallial Dc region at Page 4, Line 78.

Line 51, suggest...cortex or other parts of CNS...

Response: We revised this as suggested (Page3, Line42-43).

Line 55, suggest...or dangers and avoid detection by not moving.

Response: We revised this as suggested (Page3, Line48).

Lines 88-89 suggest: "...5-HT7 receptors. Genetic ablation or pharmacological blockade of 5-HT-7 receptors abolished the vigilance behavior".

Response: We revised this as suggested (Page4, Line81).

Line 121, should be hemispheres.

Response: We revised this as suggested (Page6, Line114).

Line 142 suggest ..discharging at a much higher frequency in... Comment .a discharge a frequency of around 3Hz is not "high frequency".

Response: We revised this as suggested (Page6, Line134).

Line156 Help the reader by defining what tph stands for.

Response: We revised this as suggested (Page7, Line147).

Line 161-164. Too brief description of the results.

Response: We have added some detailed description of these results (Page7, Line154-160).

Lines 226-229 same as line 156

Response: We revised this as suggested (Page9, Line222-223).

Line 294 should be ...simultaneous...

Response: Revised as suggested (Page12, Line295).

Line 340-341 suggest...in the pallial DC-region of zebrafish or in the prefrontal cortex...

Response: Revised as suggested (Page14, Line341).

Line 346 suggest. ...spontaneous single spike firing to.

Response: Revised as suggested (Page14, Line346-347).

Line 349 suggest: ...the neuronal populations,

Response: Revised as suggested (Page14, Line349).

Line 353 suggest:...alert..

Response: Revised as suggested (Page14, Line354).

Line 385 suggest exchange extraordinarily to much.

Response: Revised as suggested (Page16, Line390).

Line 1142 Define, “cadaverine” and difference to CAS

Response: We have added the definition of “cadaverine” in the Method (Page 18, Line 455-456).

Line 1145 Suggest: “All data here and further below will be presented as....”

Response: Revised as suggested (Page38, Line1045).

Figure 2G. The interpretation of the cross-correlation squared diagrams, control and CAS and the color code need further explanation, also since they are used on other below.

Response: We have added interpretation of the cross-correlation squared diagrams in Fig.2g (Page38, Line1062-1065).

Figure 5A expand explanation.

Response: We have expanded explanation of Fig. 5a as suggested (Page40, Line1134-1137).

Figure 6E, simplify fro the reader by labelling with 5-HT7 antagonist.

Response: We have added the information “5-HT7 antagonist” in the figure legends as suggested (Page40, Line1118-1119).

Supplementary figures:

All supplementary figures should have a title of the legend, similar to the main figures and e.g.

Response: We have added titles for all supplementary figures as suggested.

Figure S7. Perhaps some more detail in these legends.

Response: We have added some detail in these legends as suggested.

Figure S3C. Explain the procedure better and the effect of MTZ.

Response: We have expanded explanation of Supplementary Fig.3c-d as suggested.

The videos are excellent. However, the video legends could be expanded to include more detail. There is for instance a very long period before eg CAS is entered in video 1, and an impatient viewer may give up for the striking effect occurs.

Response: Thanks a lot for the suggestions. We now modify the video accordingly.

Reviewer #3 (Remarks to the Author):

The manuscript by Zhao et al uncover a conserved link between 5-HT neurons in the dorsal raphe nuclei and cortical neurons (or their homolog) underlying vigilance in zebrafish and mouse. The authors use a combination of behavioural testing, calcium imaging, electrophysiology, optogenetics and single-cell sequencing to identify a role of 5-HT in synchronizing glutamatergic neurons in the cortex through activation of 5-HT7a. The experiments are technically challenging and the results in support of their hypothesis is strong. The findings further support the role of dorsal raphe nuclei neurons in vigilance and identify the circuitry and circuit activity that lead to a vigilant state in vertebrates.

Thank you very much for the comment on our manuscript. We prepared the point-to-point response for the minor comments.

I only have minor comments:

1. Line 136. Serotonin is often used to induce locomotor activity in the isolated spinal cord. How does this observation reconcile with the reduction of locomotor activity due to DRN 5-HT neuron activity?

Response:

Thank you for the question. This could be due to activate different 5-HT neuron clusters. In this manuscript, the CAS activated the DRN 5-HT neurons, which released high concentration of 5-HT in zebrafish Dc region or mice prefrontal cortex. This resulted in the vigilance behavior state and animals displayed quiescent for better detecting the potential threat. Underlying the vigilance state, the brain might generate the halt command on the motor system and inhibit the movement. While we all know that the co-application of glutamate and 5-HT in the isolated spinal cord can induce fictive locomotion recorded on the ventral roots. The 5-HT resource in spinal cord might be from the 5-HT neurons in the ventral raphe. 5-HT neurons in ventral raphe project across spinal cord and could be the dominant resource for spinal locomotion.

Thus, it is very likely that the 5-HT neurons in DRN or VRN targeting at different brain regions play roles in halting or promoting.

2. Fig. 3R. I assume exogeneous 5-HT administration in the tph2^{-/-} line was in the absence of CAS. Is the same response seen in tph2^{+/+} line to exogeneous 5-HT (in the absence of CAS)?

Response:

Thank you for the question. Yes, exogenous application of 5-HT alone in the tph2⁺ line induced the synchronized oscillation underlying internal vigilance state in absence of CAS. We provided a figure to show that exogenous application of 5-HT alone can trigger neuronal synchrony in Dc region. Please see the figure below:

The two-photon calcium imaging showing that exogenous application of 5-HT triggered neuronal synchrony in Dc region of the HuC:H2B-GCaMP6f fishline

3. *Fig. 4C. The post-inhibitory rebound in synchronized neurons occurred at the onset of calcium oscillations yet the firing right after the PIR isn't that much higher in intensity in these neurons than in intervals between the calcium oscillations. Can you comment on how the firing after the PIR could induce calcium oscillations and synchronization but not the firing of these neurons in between calcium oscillations?*

Response:

Thank you for the great question.

First, the neurons in Dc region changed their firing pattern from tonic to bursting underlying vigilance state. However, the burst firing among these neurons never became synchronized, unless the PIR appeared. Thus, it is reluctant to say that burst firing of these neurons alone could trigger synchronized oscillation. The most important thing could be that the post-inhibitory rebound

occurred in all the bursting neuron at the same time, which immediately synchronized the activity of these neurons and generated synchronized calcium oscillation.

Second, we found that there were small calcium fluctuations in the individual neurons between the synchronized calcium oscillation. Compared with the large synchronized calcium oscillations, the amplitude of these small non-synchronized calcium fluctuations was so small to be illustrated in the figures. We now expanded a trace from Figure 2d to show these small fluctuations (please see the figure below).

The small calcium fluctuations in the baseline of the calcium analysis traces

4. Lines 459-460 How was the working concentration of CAS established?

Response:

Here is the way we prepared the CAS as we described in the methods. One adult fish was placed in a petri dish kept on ice and conspecific alarm substance (CAS) was obtained through 10-15 superficial shallow cuts on epidermal cells with a razor blade. CAS stock solution was prepared by washing on both sides of a single fish with 10 mL distilled water. To reliably induce vigilance behavior in zebrafish, 3.5 mL CAS stock solution was diluted in 1 L water as the working concentration. The working concentration can reliably trigger the vigilance behavior with a successful rate, 95%. We have listed the successful rate for a range of concentration of CAS (please see the figure below). It is clearly shown that the working concentration is the mostly reliable one.

The successful rate of vigilance behavior induced by the different concentration of CAS

REVIEWERS' COMMENTS

Reviewer #1 (Remarks to the Author):

In the revised manuscript, the authors have well addressed my concerns. I have no more comments and think it is ready for publication.

Reviewer #2 (Remarks to the Author):

The authors have responded to all items I considered in a satisfactory manner and I congratulate the authors to an interesting study.

Reviewer #3 (Remarks to the Author):

The authors have addresses my comments well. The comments from the other reviewers seem to be well addressed, and in some cases, this involved new experiments that further support the conclusions made in the paper.

Full responses to each of the points raised are given below.

Reviewer #1 (Remarks to the Author):

In the revised manuscript, the authors have well addressed my concerns. I have no more comments and think it is ready for publication.

We thank the reviewer for the comments on our manuscript.

Reviewer #2 (Remarks to the Author):

The authors have responded to all items I considered in a satisfactory manner and I congratulate the authors to an interesting study.

We thank the reviewer for the comments on our manuscript.

Reviewer #3 (Remarks to the Author):

The authors have addresses my comments well. The comments from the other reviewers seem to be well addressed, and in some cases, this involved new experiments that further support the conclusions made in the paper.

We thank the reviewer for the comments on our manuscript.